# Zebularine potentiates anti-tumor immunity by inducing tumor immunogenicity and improving antigen processing through cGAS-STING pathway

Yong Zhang [1,2], Heng Zhao[1,2], Weili Deng[1,2], Junzhong Lai[3], Kai Sang[1,2] & Qi Chen [1,2] ✉

DNA methylation is an important epigenetic mechanism involved in the anti-tumor immune response, and DNA methyltransferase inhibitors (DNMTi) have achieved impressive therapeutic outcomes in patients with certain cancer types. However, it is unclear how inhibition of DNA methylation bridges the innate and adaptive immune responses to inhibit tumor growth. Here, we report that DNMTi zebularine reconstructs tumor immunogenicity, in turn promote dendritic cell maturation, antigen-presenting cell activity, tumor cell phagocytosis by APCs, and efficient T cell priming. Further in vivo and in vitro analyses reveal that zebularine stimulates cGAS-STING-NF-κB/IFNβ signaling to enhance tumor cell immunogenicity and upregulate antigen processing and presentation machinery (AgPPM), which promotes effective CD4$^+$ and CD8$^+$ T cell-mediated killing of tumor cells. These findings support the use of combination regimens that include DNMTi and immunotherapy for cancer treatment.

Cancer incidence and mortality are rising rapidly worldwide[1,2]. Recent advances in cancer therapeutics have leveraged the activation of anti-tumor immune responses to eliminate cancer cells. The benefits of anti-cancer immunotherapies, including checkpoint inhibitors, recombinant cytokines, and chimeric antigen receptor T cell therapy, have been demonstrated in many cancer types; however, only a small percentage of tumors respond to immunotherapy. The innate immune system is the first line of defense against pathogens and malignancies[3,4], and its activation is an important factor in the effectiveness of cancer treatment[5,6]. The adaptive immune response against cancer depends on (1) recognition of neo-antigens on tumor cells by naive T cells[7–9] and (2) delivery of adjuvant-like "danger" signals by tumor cells to antigen-presenting cells (APCs) in the form of exogenous microbe-associated molecular patterns (MAMPs) or endogenous damage-associated molecular patterns (DAMPs)[10,11]. Notably, most DAMPs or MAMPs mediate the cross-presentation of tumor antigens to CD8$^+$ cytotoxic T lymphocytes (CTLs) through pattern recognition receptors (PRRs) on APCs, including dendritic cells (DCs), macrophages, and other components of the innate immune system, ultimately promoting the recognition and elimination of cancer cells[12].

Some chemotherapy drugs are not only directly cytotoxic, but also alter the immunogenicity of tumor cells[13–15]. In particular, immunogenic cell death (ICD), a form of regulated cell death, plays a critical role in chemotherapeutic efficacy. ICD is accompanied by the exposure and release of DAMPs or MAMPs, translocation of the ER chaperone calreticulin (Calr) to the plasma membrane, ATP secretion to the extracellular space, and HMGB1 leakage from the nucleus to the cytoplasm[16]. In addition, ICD is associated with secretion of interferon β (IFNβ) and Cxcl10[12,17,18]. In ICD, the dying tumor cells act as endogenous vaccines or adjuvant signals that attract immune cells into the tumor microenvironment (TME). Antigenic peptides on the surface of tumor cells are taken up, processed, and presented by APCs, which in turn activate T cells to attack the tumor cells[6,19]. Hence, these DAMPs or MAMPs may influence adaptive immune responses, particularly crosspriming. In this way, ICD induced by chemotherapy drugs activates anti-cancer immunity and enhances the efficacy of immunotherapies.

In recent decades, DNA methylation inhibitors (DNMTi) has been widely used in tumor therapy[20–26]. The DNMTi upregulates MHC-I in tumor cells promoting recruitment of CD8$^+$ T cells to the microenvironment, but little is known about its interplay with the innate immune system. Here, we found that DNMTi induce immunogenic cell death in tumor cells, significantly promotes the maturation of DCs, increases tumor cell phagocytosis by APCs, and enhances T cell priming. In addition, the anti-tumor

[1]Fujian Key Laboratory of Innate Immune Biology, Biomedical Research Center of South China, Fujian Normal University Qishan Campus, Fuzhou, Fujian Province 350117, China. [2]College of Life Science, Fujian Normal University Qishan Campus, Fuzhou, Fujian Province 350117, China. [3]The Cancer Center, Union Hospital, Fujian Medical University, Fuzhou, Fujian Province 350117, China. ✉e-mail: chenqi@fjnu.edu.cn

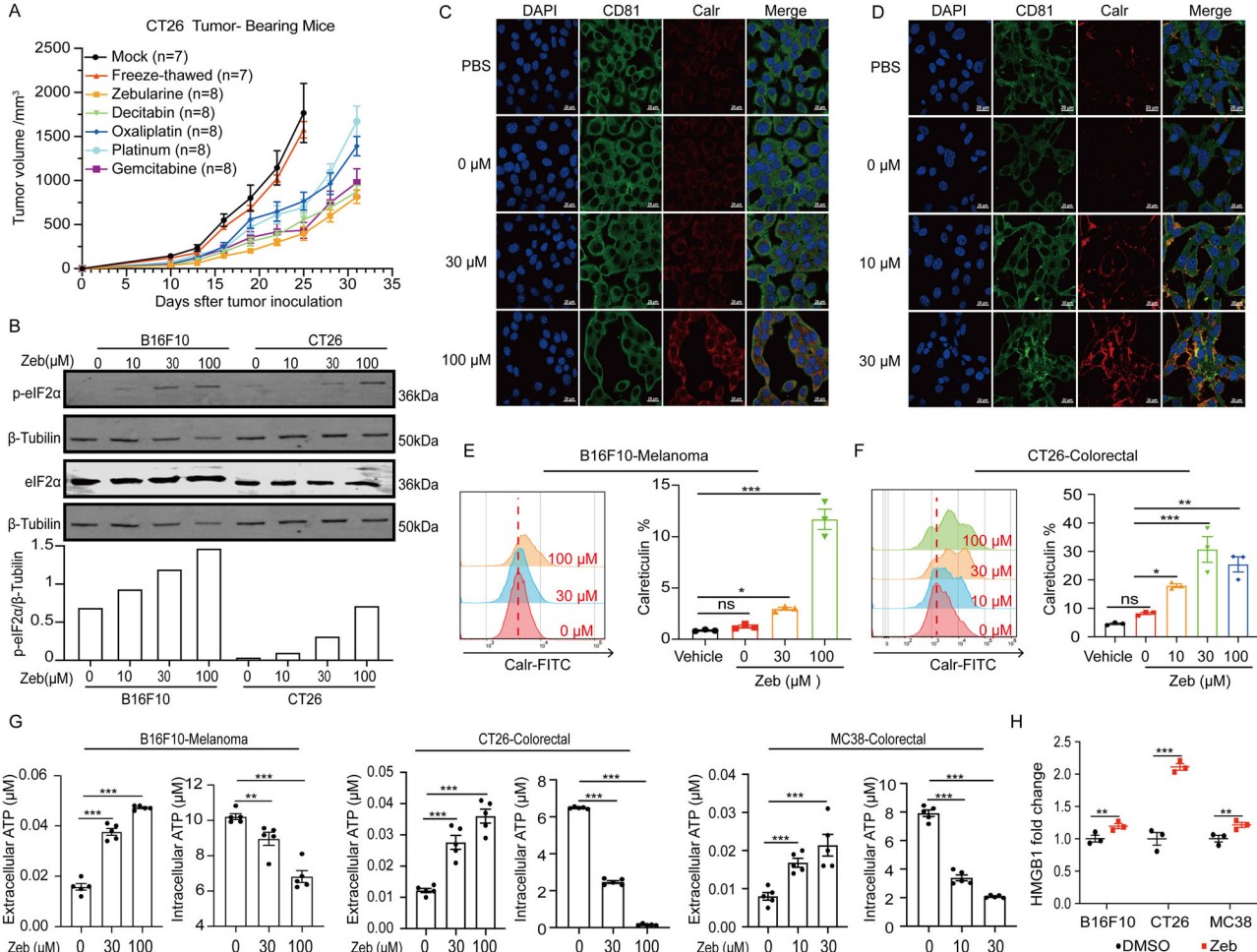

**Fig. 1 | DNMTi-induced immunogenic cell death in tumor cells. A** Vaccination experiments showing changes in tumor volume. CT26 tumor cells were pre-treated with zebularine (zeb, 150 µM), decitabin (100 µM), oxaliplatin (100 µM), platinum (30 µM), or gemcitabine (10 µM) for 3 days followed by subcutaneous inoculation into BALB/c mice as a vaccine, freeze-thawed as negative control. After 8 days, mice were re-challenged with live CT26 cells, and live CT26 cells were also implanted into non-immunized mice ($n = 7 \sim 8$) as a control. **B** Immunoblot analysis of WCL (whole cell lysates) derived from B16F10 and CT26 cells treated with different concentrations of zeb for 72 h (Uncropped blots show Supplementary Fig. S11).

**C, D** Calr (red) translocation in B16F10 (**C**) and CT26 (**D**) cell lines after treatment with various concentrations of zebularine, observed by confocal microscopy. CD81 (green) is a cell membrane marker. **E, F** Calr translocation from the ER lumen to the cell membrane surface in B16F10 (**E**) and CT26 (**F**) cells after treatment with various concentrations of zebularine, was observed by flow cytometry. **G** Extracellular and intracellular ATP content analyzed by luciferase assay. **H** HMGB1 release into supernatant determined by ELISA. Data are presented as mean ± SEM of at least three independent experiments. *$p < 0.05$; **$p < 0.01$; ***$p < 0.001$ by Student's *t*-test.

effects of DNMTi -zebularine depend on the co-existence of CD4$^+$ and CD8$^+$ T lymphocytes and the presence of CD4$^+$ T cells is necessary for CTLs to kill tumor cells. Moreover, we revealed that cGAS-STING–induced activation of NF-κB and IFNβ signaling drives the innate immune response to DNMTi and further activates MHC (major histocompatibility complex) antigen processing and presentation machinery (AgPPM) to initiate more effective CD4$^+$ and CD8$^+$ T cell killing of tumor cells.

## Results
### DNMTi-induced immunogenic cell death in tumor cells
Accumulating evidence has shown that the capacity of anti-cancer agents to induce ICD is positively correlated with their therapeutic effect[18,27–31]. However, most anti-cancer agents eliminate tumor cells by inducing non-immunogenic cell death, and only a handful of agents are recognized as bona fide ICD inducers, including dinaciclib, oxaliplatin, and gemcitabine[32,33]. We previously showed that the DNMTi zebularine promotes infiltration of CD8$^+$ T cells and NK cells into tumors. Here, we asked if DNMTi bridge the innate and adaptive responses to inhibit tumor growth. In vaccination experiments, DNMTi treatment with either zebularine or decitabine, killed tumor cells and primed anti-tumor immunity (Fig. 1A and Supplementary

Fig. 1a, b). In our previous study, we found that treatment of the gastric cancer cell line AGS with zebularine for 4 days promoted low-level upregulation of *IFNβ* and *CXCL10* gene expression in a dose-dependent manner[34]. We observed this phenomenon in a variety of human and mouse tumor cell lines (Supplementary Fig. 1c). Furthermore, e-IF2α phosphorylation was upregulated in a dose-dependent manner in B16F10 and CT26 cell lines (Fig. 1B). PERK-dependent phosphorylation of e-IF2α has been shown to facilitate the translocation of Calr from the ER lumen to the cell membrane surface[35], and recent studies have shown that anti-cancer drugs promote the upregulation of e-IF2α phosphorylation and *IFNβ* and *Cxcl10* expression in tumor cells, which may lead to ICD[32,33,35]. Thus, we posited that DNMTi zebularine might enhance anti-tumor immunity by inducing ICD in tumor cells.

To test this hypothesis, we first examined changes in several ICD biomarkers following zebularine treatment, including Calr translocation, HMGB1 release from the nucleus, and ATP level. As shown by confocal microscopy (Fig. 1C, D), when B16F10 (Fig. 1C), CT26 (Fig. 1D) cells were treated with zebularine, we observed a significant increase in the translocation of Calr from the ER lumen to the cell membrane surface in a dose-dependent manner. We also found an increase in Calr on the cell membrane after zebularine treatment of B16F10 (Fig. 1E) and CT26 (Fig. 1F) cells, as

analyzed by flow cytometry. Moreover, when B16F10, CT26, and MC38 cells were treated with increasing concentrations of zebularine for 3 days, we observed ATP release from tumor cells into the extracellular media in a dose-dependent manner (Fig. 1G) as well as HMGB1 leaking from the nucleus into the cytoplasm by confocal microscopy (Supplementary Fig. 1d, e). HMGB1 in the medium was detected at extremely low levels (Fig. 1H). These in vitro data indicating that zebularine promotes the upregulation of ICD-related biomarkers in tumor cells—together with the gold standard evidence of zebularine-induced ICD in vaccination experiments as shown in (Fig. 1A and Supplementary Fig. 1a)[32,36,37]—confirm the immunogenic properties of tumor cell death induced by zebularine.

## Activation of immunogenicity enhances APC maturation and phagocytosis

The expression of Calr on the cell membrane is upregulated during ICD in tumor cells and is recognized by APCs including DCs and macrophages as a signal for phagocytosis. As shown in Fig. 2a, we found that expression of Calr on the cell membrane was upregulated by zebularine. Also during ICD, the release of HMGB1, IFNβ, and ATP from tumor cells further stimulates the maturation of DCs[11,27]. Therefore, we pre-treated tumor cells with zebularine for 3 days followed by co-culture with immature bone marrow-derived dendritic cells (BMDCs). Mature BMDCs were detected by flow cytometry (Fig. 2b–d). We found that pre-treatment with zebularine significantly promoted the expression of CD80+ and CD86+, surface markers of BMDC activation, compared with the untreated control group (Fig. 2c, d and Supplementary Fig. 2a). These results indicate that exposure of tumor cells to zebularine promoted the maturation of BMDCs. Innate immune cells within draining lymph nodes (dLNs) also contribute to the induction of T cell responses[38,39]; thus, we evaluated the maturation of DCs in the TME and dLNs in B16F10 and MC38 tumor-bearing mice treated with zebularine. Compared with the control group, tumors from the zebularine-treated group showed greater infiltration of mature DCs (Fig. 2e, f) and significantly more mature DCs in dLNs (Fig. 2g, h and Supplementary Fig. 2d, e and Supplementary Fig. 8a) and CD4+ and CD8+ T cells in the TME (Fig. 2i). As seen in Fig. 2j, a greater extent of macrophage infiltration was found in B16F10 tumor-bearing mice treated with zebularine. These results indicate that ICD induced by zebularine promotes the maturation of BMDCs and the infiltration of DCs, T cells, and macrophages into the TME.

As mentioned above, ICD stimulates expression of Calr on the cell surface of tumor cells, which promotes their recognition and phagocytosis by APCs. We next investigated whether pre-treatment of tumor cells with zebularine enhances their phagocytosis (Fig. 2k and Supplementary Fig. 2b). Pre-treatment of tumor cells with zebularine significantly increased phagocytosis of B16F10 and CT26 cells by APCs compared to the control group (Fig. 2l–o). Tumor ICD-promoted mature DCs can present antigens to naïve T cells, further promoting their differentiation into CD8 or CD4 and ultimately eliciting a T-cell immune response[40,41], we next investigated whether co-culture of zebularine-pretreated tumor cells, BMDCs, and mouse splenic lymphocytes more effectively promotes T cell priming (Fig. 2p). As shown in Fig. 2q, r, when tumor cells were pre-treated with zebularine, both CD4+ and CD8+ T cells in the co-incubation system proliferated significantly and were more abundant than those in control groups (Fig. 2q, r). Thus, these results indicate that zebularine enhances the cross-presentation of tumor antigens by promoting tumor cell phagocytosis, thereby enhancing T cell activation.

## Zebularine affects tumor growth in a T cell-dependent manner

Because zebularine promoted the maturation and phagocytosis of APCs and facilitated the initiation of T cells, we next asked whether zebularine-induced ICD relies on a functional immune system. Zebularine treatment of both immunocompetent (Fig. 3a) and immunocompromised mice (Fig. 3b) was initiated once tumors reached 60–80 mm³ in volume. We observed that zebularine treatment delayed tumor growth and reduced tumor weight only in mice with an intact immune system (Fig. 3a, b). We further found that spleen T lymphocytes isolated from zebularine-treated mice significantly

enhanced T cell killing of tumors (Supplementary Fig. 2c and Supplementary Fig. 3a, c) and increased IFNγ secretion (Supplementary Fig. 3d, e). These results indicate that zebularine enhances T lymphocyte cytotoxicity through immunomodulatory effects.

To explore the contribution of T cells to the anti-tumor effect of zebularine, we used monoclonal antibodies (mAb) to target CD4+ or CD8+ T cells in zebularine-treated tumor-bearing immunocompetent C57BL/6 mice. We observed that the anti-tumor effect of zebularine was abolished in the presence of anti-CD8 or anti-CD4 antibodies (Supplementary Fig. 3h) in B16F10 tumor-bearing mice (Fig. 3c, d, Supplementary Fig. 3i and Supplementary Fig. 3k) or MC38 tumor-bearing mice (Fig. 3e and Supplementary Fig. 3j and Supplementary Fig. 3l). Next, we injected B16F10 cells subcutaneously into syngeneic mice; when the tumor volume reached at least 60–80 mm³, the mice were treated with zebularine or PBS for 12 days, after which spleen CD3+ T lymphocytes were isolated (Fig. 3f). Spleen CD3+ T cells were then injected via the tail vein into syngeneic (Fig. 3g) or NCG mice (Fig. 3h) when the tumor volume reached ~60–80 mm³. Interestingly, we observed that only the CD3+ T lymphocytes derived from zebularine-treated mice inhibited tumor growth (Fig. 3g, h and Supplementary Fig. 3f and Supplementary Fig. 3g), whereas adoptive therapy with CD4+ or CD8+ T cells alone did not delay tumor growth (Fig. 3i, j). In summary, these results suggest that the anti-tumor effects of zebularine depend on the co-existence of CD4+ and CD8+ T lymphocytes and the presence of CD4+ T cells is necessary for CTLs to kill tumor cells.

## Distinct functional signatures of the melanoma tumor cells

To further evaluate the effect of zebularine-induced ICD on tumor growth inhibition in tumor-bearing immunocompetent mice, we performed single-cell RNA sequencing of CD45- cells in the tumor tissue from tumor-bearing mice after 12 days of zebularine treatments. We obtained 16862 cells and performed unsupervised t-SNE dimensionality reduction analyses on the sequencing data[42,43]. Clustering was performed using the Seurat package and further annotated for known cell type-specific markers[44]. These analyses suggested that these cells were made up mostly by melanoma tumor cells (Dct, Pmel, Tyrp1, Mlana), and a small percentage of T cells (Cd2, Cd3e, Cd3g), CAF (Col1al, Col3al, Col5al, Fstl1, Bgn), and TAM cells (Cd68, Itgam, Csf1r, Adgre1, Cd14, C1qa) (Supplementary Fig. 4a, b). Since we used CD45- magnetic beads for sorting; Therefore, a small number of immune cells are present after subpopulation annotation analysis. Thus, we will analyze the melanoma cells.

We grouped melanoma tumor cells into 7 main clusters based on enrichment of high expressed genes, and found that these 6 clusters were characterized by distinct functional signatures based on Gene Ontology analysis (Fig. 4a, b and Supplementary Fig. 4c, d). The B16-Tpm1 cluster was featured by enrichment of cytoskeleton regulation and cell adhesion. The B16-H3c13 cluster was involved in chromatin organization and histone methylation. The B16-Cxcl10 cluster highly expressed Ifit3b, Tgtp1, Ifit1 and Isg15 mRNAs, which are involved in the immune-related functions such as interferon response, Toll like receptor signaling, and chemokine signaling. The B16-Ccna2 cluster involved highly expressed genes such as Cenpf, Ccna2, Cdca8 and Cdk1, which are associated with cell cycle and mitosis functions. Further cell cycle analysis of cancer cells showed that zebularine inhibited tumor cell growth, mainly by suppressing cell cycle G1 phase and expression of cell proliferation-related genes Pcna; however, we did not see the consistent change in the mRNA expression of Mcm6 and Mki67 by zebularine (Fig. 4c, d and Supplementary Fig. 5a). Interestingly, IHC staining showed that both Pcna and Mki67 expression were significantly reduced by zebularine at the protein level compared with controls (Fig. 4e, f). How zebularine regulates the down-regulation of Mki67 protein expression needs to be further investigated. Thus, we suggest that zebularine inhibits the growth of tumor cells by affecting the cell cycle and cell proliferation. The B16-Cxcl10 and -Ccna2 clusters have been described by the previous reports[45,46]. The B16-Gsta2 cluster was enriched for metabolism of xenobiotics by cytochrome P450, response to reactive oxygen species, and fatty acid metabolism; In addition, a subsets of genes function in the same way as

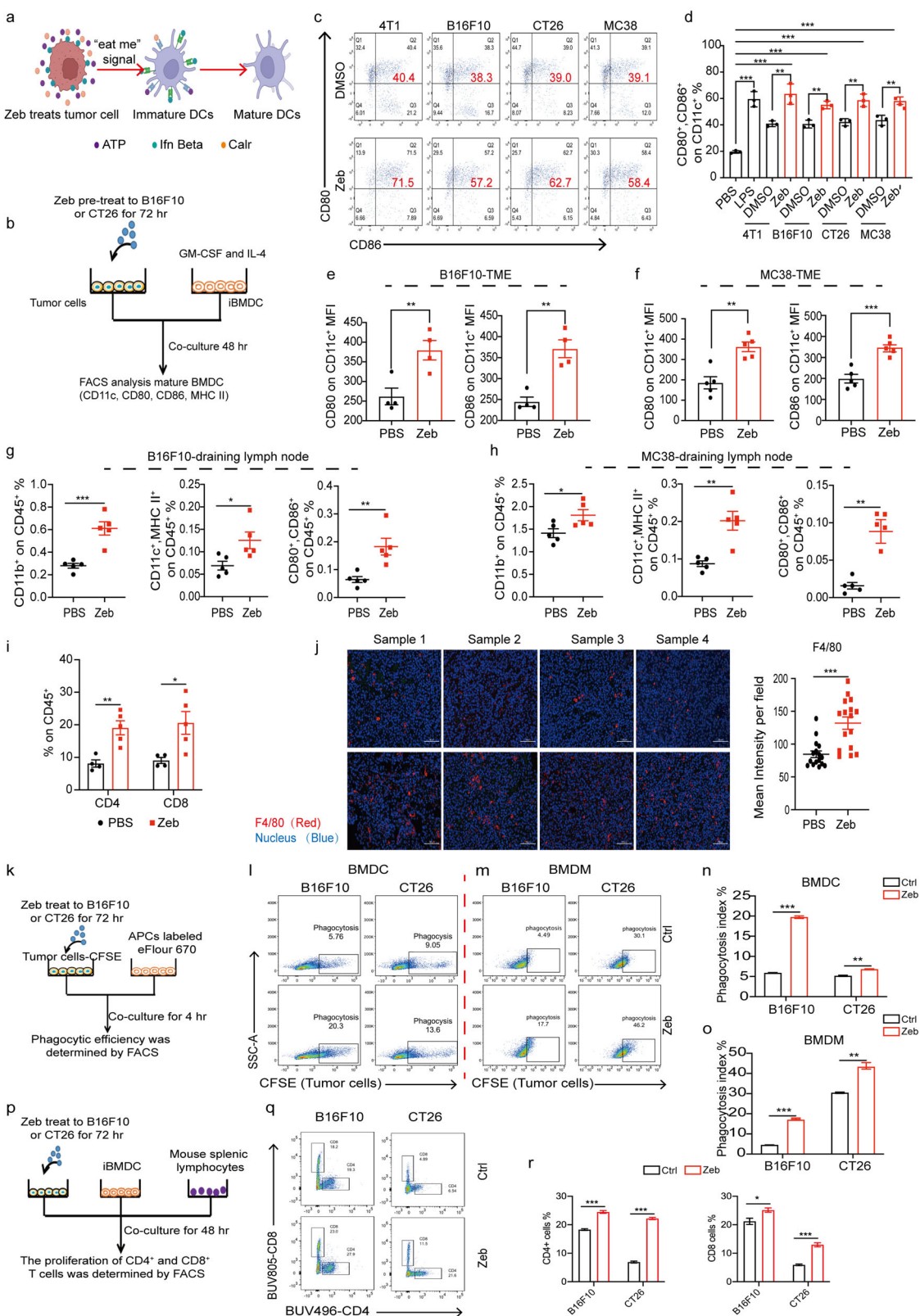

the B16-Cxcl10 cluster (such as interferon response and TNF signaling). The B16-Vegfa cluster was associated with VEGF signaling pathway and galactose metabolism. From these clusters, we found that the number of B16-Ccna2, B16-Cxcl10 and B16-Tpm1 clusters was increased, and the number of B16-Vegfa, B16-Gsta2 clusters was decreased in the zebularine group compared with the mock group (Supplementary Fig. 4a). These

results indicated that zebularine treated tumor cells may promote inflammatory response, inhibit cell proliferation and reduce cell energy metabolism (Supplementary Fig. 4a). The above data suggest that the development of melanoma could be generally related to angiogenesis, cell cycle, inflammation and energy metabolism, which further characterizes the heterogeneity of cells in melanoma.

**Fig. 2 | Activation of immunogenicity enhances APC maturation and phagocytosis. a** Schematic of tumor cell immunogenic death promoting dendritic cell (DC) maturation. **b** Schematic of co-culture of tumor cells and immature BMDCs. **c, d** Flow cytometric analysis showing expression of CD80+ and CD86+, the main markers of DC maturation, after 48 h of co-culture of zebularine pre-treated tumor cells and immature BMDCs. **e, f** Flow cytometry to detect expression of CD80 and CD86 dendritic cells in the tumor microenvironment (TME) of tumor-bearing mice after 12 days of zebularine treatment. **g, h** Flow cytometry to detect the expression of CD11b+, CD11c+ MHCII+, and CD80+ CD86+ cells in draining lymph nodes of tumor-bearing mice treated with zebularine for 12 days. **i** Flow cytometry to detect the expression of CD4+ CD45+ and CD8+ CD45+ T cells in the TME of B16F10 tumor-bearing mice after 12 days of zebularine treatment. **j** Immunofluorescence staining for monocytes/macrophages (F4/80, Red) and nuclei (DAPI, Blue) in

B16F10 tumor-bearing wild-type mice treated with zebularine for 12 days. The number of monocyte/macrophages in tumors was quantified based on F4/80-Red intensity per field using ZEISS software. **k** Schematic of APC (BMDC or BMDM) phagocytosis of tumor cells in the binary co-incubation system. eFlour670 Red-labeled BMDC or BMDM were co-cultured with CFSE-stained B16F10 or CT26 cells for 4 h and then analyzed by flow cytometry. **l–o** Representative flow cytometry plots (**l, m**). Phagocytosis, calculated as the percentage of the total number of BMDCs (**n**) or BMDMs (**o**) containing cancer cells. **p** Schematic of BMDCs, tumor cells, and splenic T cells in the ternary co-incubation system, analyzed by flow cytometry. **q–r** Flow cytometry analysis of percentages of CD4+ and CD8+ T cells in the ternary co-incubation system. Data are presented as mean ± SEM of at least three independent experiments. *$p < 0.05$; **$p < 0.01$; ***$p < 0.001$ by Student's t-test.

## Activation of immunogenicity enhances a variety of immune response genes and promotes anti-tumor immunity

To further investigate the physiological effects of zebularine-induced ICD in tumor cells, we performed differentially expressed genes (DEGs) analysis on the single-cell transcriptome data and found that *Ccl5*, *B2m*, *H2-D1*, *Cxcl10*, *Isg15*, *Psmb9*, and *Psmb8* were significantly upregulated in the zebularine group compared with the control untreated group (Fig. 5a). Interestingly, gene ontology (GO) and KEGG analyses showed that the top 20 DEGs were enriched in biological processes associated with antigen processing and presentation and the innate immune response (Fig. 5b, c). In addition, we also found that several IFN-sensitive transcription factors (e.g., *Stat1*, *Stat2*), IFN-stimulated genes (e.g., *Isg15*, *Oas1a*, *Oas2*, *Ifit1*, *Ifit3*, *Bst2*), and IFN-inducible T-cell chemokines (e.g., *Cxcl10*, *Ccl2*, *Ccl7*) were all significantly upregulated in the zebularine-treated group (Fig. 5d). Upregulation of these genes would be expected to improve the infiltration of T cells into tumors, allowing T cells to kill tumor cells and inhibit tumor growth. Notably, we used reverse transcriptional quantitative PCR (RT-qPCR) to confirm that the genes involved in IFN and viral responses were significantly upregulated in B16F10-melanoma cell lines treated with zebularine (Fig. 5e). We observed a similar result in MC38-colorectal cell lines (Fig. 5f) and in B16F10 tumor-bearing mice (Fig. 5g). These data suggest that zebularine-induced tumor immunogenicity may lead to anti-tumor immunity by enhancing tumor-specific T cell activity triggered by IFN and upregulation of T-cell chemokine-related genes.

## DNMTi upregulates antigen presentation and processing

We further analyzed our single-cell RNA-seq data and found that some genes involved in MHC mediated antigen processing and presentation were significantly upregulated in the zebularine group compared with the control group, including *B2m*, *Tap1*, *Tap2*, *Psmb8*, and *Psmb9* (Figs. 5a and 6a). Notably, RT-qPCR data confirmed significantly elevated expression of these antigen processing and presentation pathway-associated genes in zebularine-treated B16F10-melanoma (Fig. 6b), MC38-colorectal (Fig. 6c), and A375-melanoma (Fig. 6d) tumor cell lines. Moreover, similar expression patterns were seen in B16F10 tumor-bearing mice (Fig. 6e). We also observed by flow cytometry that zebularine promoted the upregulation of tumor cell surface antigen H-2Kb/H-2Db in B16F10-melanoma (Fig. 6f, g) and MC38-colorectal (Fig. 6h, i) cell lines in a dose-dependent manner, as well as HLA-A/B/C expression in the A375-melanoma cell line (Fig. 6j). We further found that small interfering RNA (siRNA) knockdown of the *Dnmt1* gene encoding DNA methyltransferase promotes upregulation of the tumor cell surface antigen H-2Kb/H-2Db in B16F10 cells (Fig. 6k, l). These results indicate that inhibition of DNA methylation, either by knocking down DNA methylation transferase expression or by treatment with zebularine, promotes the processing and presentation of tumor cell antigens.

## DNMTi-induced MHC-AgPPM gene expression is dependent on cGAS-STING pathway

We next studied how DNMTi zebularine regulates the processing and presentation of tumor cell antigens. Our STRING[47] analyses revealed interactions among MHC-AgPPM, proteasomal, and IFN-related

pathways (Supplementary Fig. 10a). We hypothesized that zebularine-induced upregulation of genes involved in tumor antigen processing and presentation might be correlated with its promotion of IFN and chemokine expression. As shown in Fig. 5c, KEGG pathway enrichment analysis data showed differential expression of genes related to the NOD-like receptor signaling pathway, Toll-like receptor signaling pathway, and cytoplasmic DNA -sensing pathway (Fig. 5c). Of note, it was previously reported that activated NF-κB can promote upregulation of MHC AgPPM gene expression[48]. Indeed, zebularine treatment of B16F10-melanoma (Supplementary Fig. 6a) and A375-melanoma (Supplementary Fig. 6b) cell lines promoted the nuclear translocation of the NF-κB subunit Rela/P65. To determine the contribution of NF-κB signaling to tumor antigen processing and presentationin response to zebularine treatment, we generated Rela/p65-deficient B16F10 cell lines (Fig. 7a). Ablation of Rela/p65 abrogated zebularine-induced expression of surface H-2Kb/H-2Db and MHC-AgPPM genes (*B2m*, *Psme1*, *Psmab8*, and *Psmb9*; Fig. 7b, c). We observed a similar effect when we treated B16F10 cells with zebularine in conjunction with a selective NF-κB inhibitor, BMS-345541 (Fig. 7d).

Our previous study found that the anti-tumor effect of zebularine was completely abrogated in *Cgas*−/− and *Sting*gt/gt mouse tumor models[34], and that NF-κB is activated in the STING signalosome[49,50]. Therefore, we hypothesized that zebularine relies on cGAS-STING signaling to facilitate antigen processing and presentation, thereby enhancing cancer cell clearance by tumor-specific cytotoxic T cells. To test this hypothesis, we generated *Cgas*−/−, *Sting*−/−, *Tbk1*−/−, and *Irf3*−/− -deficient B16F10 cell lines (Supplementary Fig. 7a, b). We found that cGAS-STING was necessary for full induction of *B2m*, *Tap1*, *Psmab8*, and *Psmb9* expression in response to zebularine both in vitro (Supplementary Fig. 7e) and in vivo (Supplementary Fig. 7f). Consistently, ablation of the cGAS-STING signaling pathway strongly inhibited surface H-2Kb/H-2Db induction by zebularine (Fig. 7e, f and Supplementary Fig. 8b, c). The same effect was seen in decitabine-treated cells (Supplementary Fig. 7d), suggesting that DNMTi-induced antigen processing and presentation relies on cGAS-STING signaling.

IFNβ is one of the most important effectors downstream of the cGAS-STING pathway. In *IFNAR1*−/− mice tumor models, we found that *IFNAR1* depletion abrogated the anti-tumor effect of zebularine (Fig. 8b). In addition, when B16F10 cells were treated with different concentrations of IFNβ alone or zebularine combined with IFNβ for 2 days, RT-qPCR revealed upregulation of MHC-dependent expression of genes involved in antigen processing and presentation (Supplementary Fig. 7g). Moreover, we used B16F10 tumor-bearing *IFNAR1*−/− mice to examine zebularine-dependent type I interferon signaling in promoting the expression of genes involved in tumor antigen processing and presentation. Interestingly, we found that zebularine-induced expression of MHC-AgPPM genes was abolished in *IFNAR1*−/− mice compared to wild-type mice (Fig. 7g). It has been reported that enhanced expression of MHC-I and the antigen presentation complex is primarily dependent on IFNγ[51]. We confirmed that IFNγ promotes the upregulation of MHC-AgPPM genes to a greater extent than the same concentration of IFNβ (Supplementary Fig. 7g). Thus, our results suggest that promotion of tumor antigen processing and presentation by zebularine

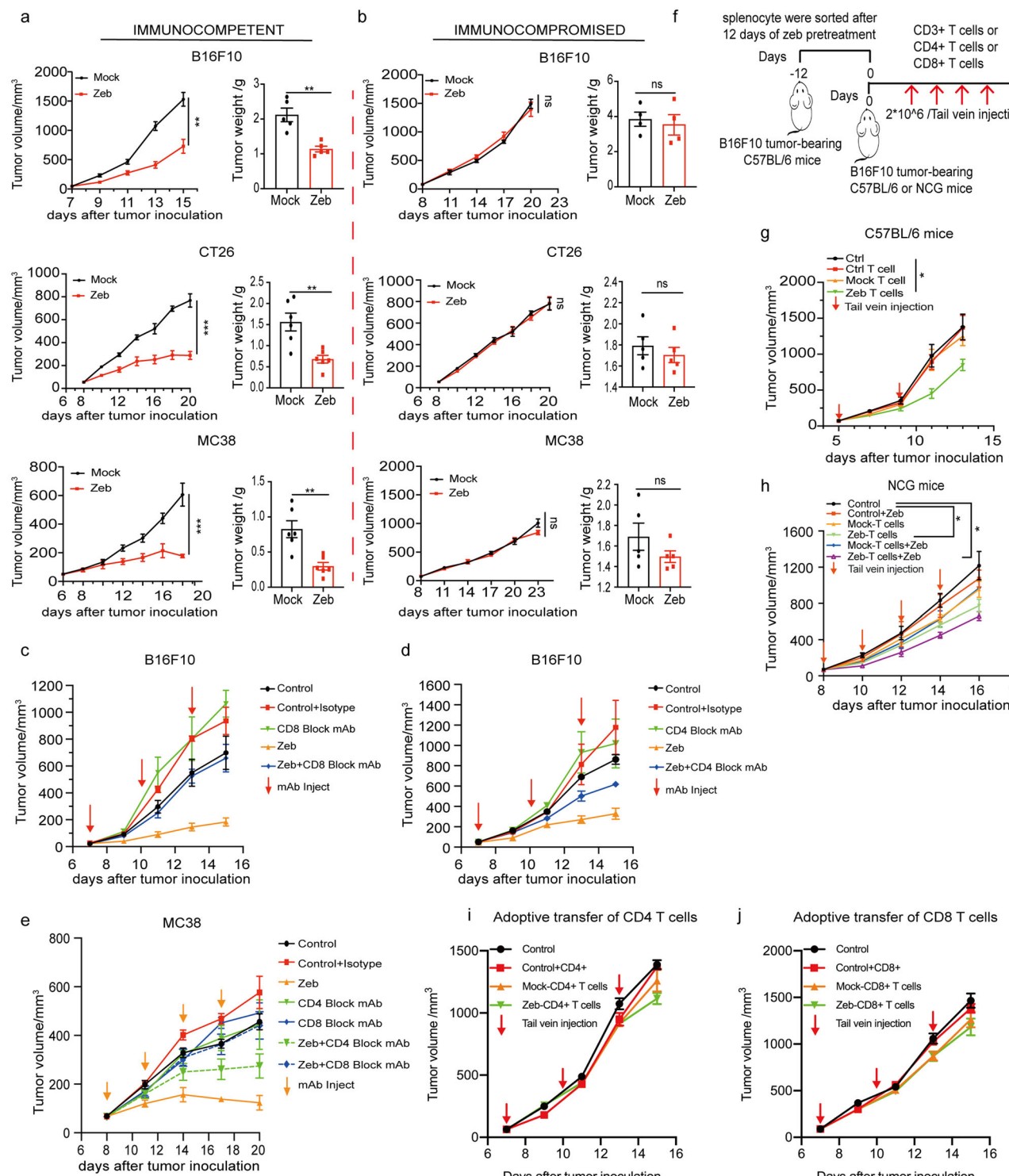

**Fig. 3 | Zebularine affects tumor growth in a T cell-dependent manner.**
**a** Immunocompetent syngeneic mice (C57BL/6 or BALB/c) were subcutaneously injected with $1 \times 10^6$ B16F10, CT26, or MC38 cells. Mice ($n = 6$) were treated with zebularine when tumor volume reached 60–80 mm³. **b** In parallel, mouse tumor cells were injected into immunocompromised NOD-SCID mice ($n = 4 \sim 5$), and treatment was conducted as in (**a**). **c, d** Depletion of CD8 T cells (**c**) and CD4 T cells (**d**) in the B16F10 tumor models. Mice ($n = 4 \sim 5$) were treated with anti-CD8 or anti-CD4 depleting mAb (200 μg per mouse, indicated by the arrows). Control mice were administered the isotype antibody. **e** Depletion of CD8 T cells and CD4 T cells in the MC38 tumor model ($n = 5$), as described in (**c**) and (**d**). **f** Schematic of adoptive T cell

transfer experiment. **g** CD3+ T cells isolated from the spleens of immunocompetent mice after 12 days of zebularine treatment or untreated control were injected into B16F10 tumor-bearing mice via tail vein; ctrl-T cell indicates tumor-bearing tumor mice injected with untreated control T cells ($n = 6$). **h** As in (**g**), untreated or zebularine-treated CD3+ T cells were injected into B16F10 tumor-bearing NOD-SCID mice ($n = 4$) via tail vein, then the mice were untreated or treated with additional zebularine. **i, j** As in (**g**), zebularine-treated or untreated CD4+ T cells (**i**) or CD8+ T cells (**j**) cells were injected into B16F10 tumor-bearing mice ($n = 4$). Data are presented as mean ± SEM of at least three independent experiments. *$p < 0.05$; **$p < 0.01$; ***$p < 0.001$ by Student's $t$-test.

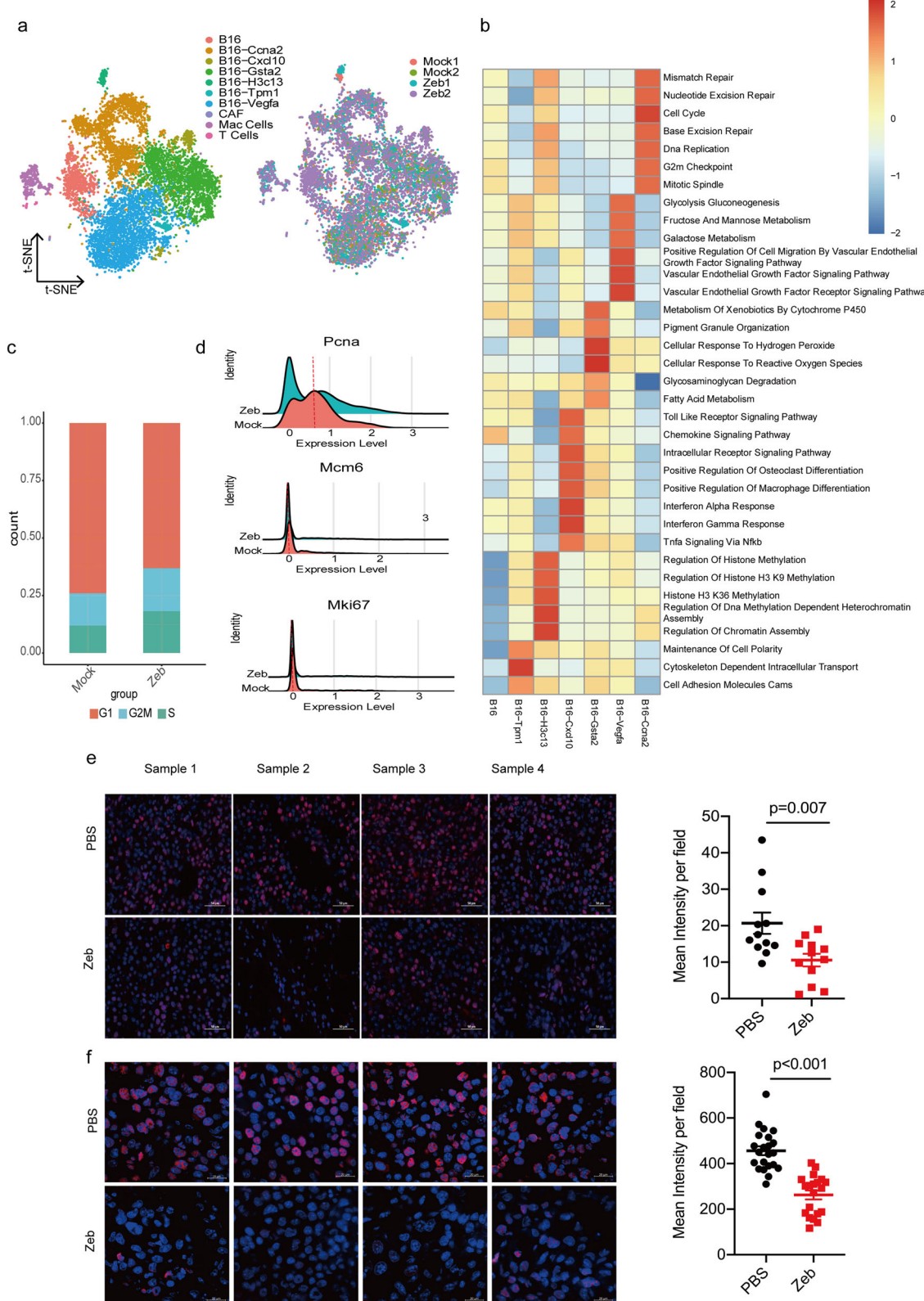

**Fig. 4 | Distinct functional signatures of the melanoma tumor cells.**
**a** T-distributed stochastic neighbor embedding (*t*-SNE) plot, showing the annotation and color codes for mock and zebularine-treated B16F10 melanoma and their integration. **b** Heat map showing the expression score by ssGESA in the 7 clusters, including related signal pathways. **c, d** Histograms represent the proportion of cell cycles and the expression levels of related genes in B16F10 after zebularine treatment. **e-f** B16F10 tumor-bearing wild-type mice were treated with zebularine. After 12 days, tumor tissue was collected. Immunofluorescence images showing proliferation of B16F10 tumor tissues, based on Mki67 (**e**) and Pcna (**f**), in the mock and zebularine groups. Data are presented as mean ± SEM of at least three independent experiments. *$p < 0.05$; **$p < 0.01$; ***$p < 0.001$ by Student's *t*-test.

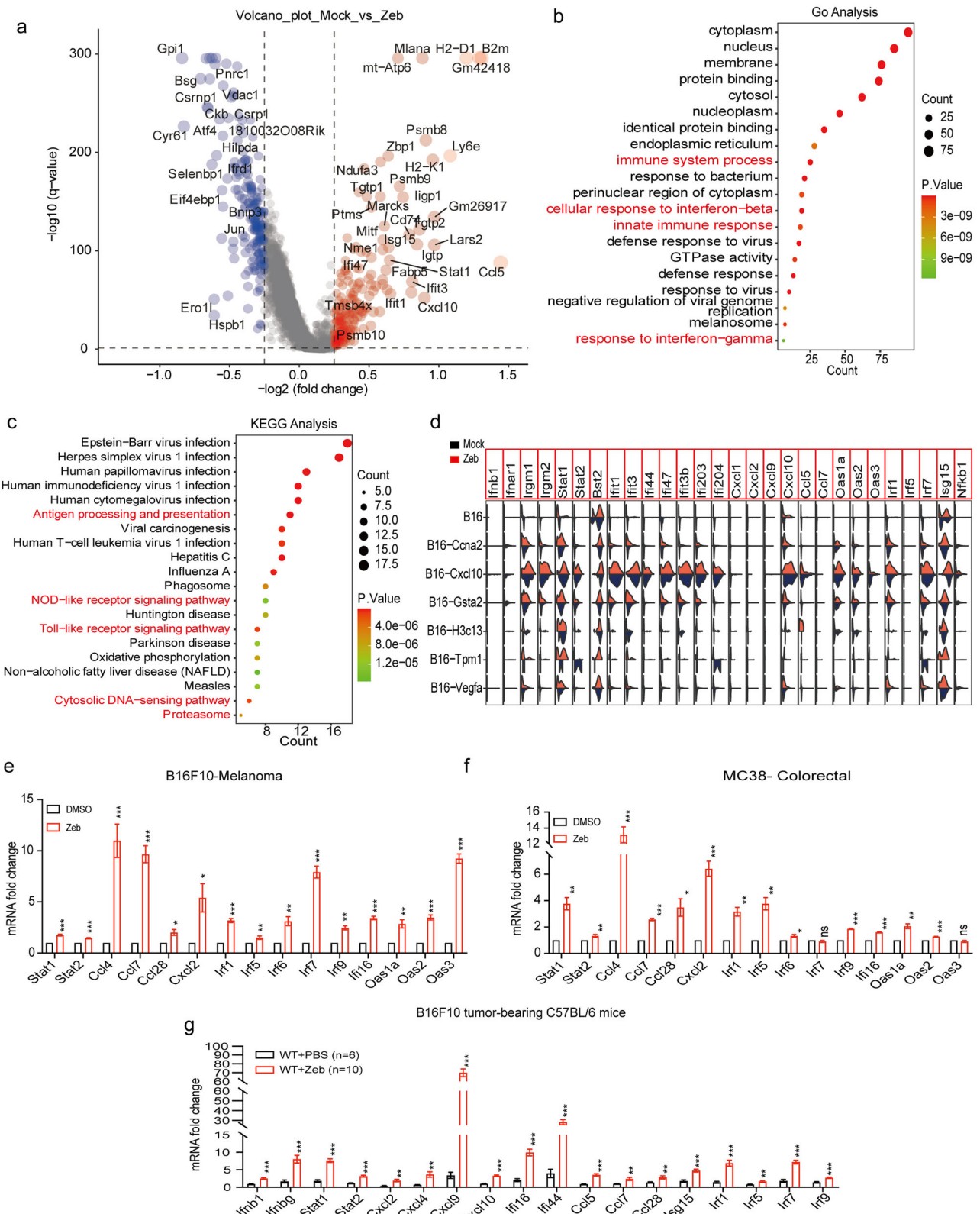

**Fig. 5 | Single cell analysis identified intracatumoral cell populations and confirmed the association of zebularine in tumor innate signaling and immunity. a** Volcano plot of single-cell RNA-seq data shows differentially expressed genes (DEGs) in mock (blue dots) and zeb (red dots) group. The most significant DEGs are indicated on the plot. **b** Gene Ontology (GO) shows the enrichment of upregulated genes. **c** KEGG pathway analysis shows the pathways of the enriched by upregulated genes. **d** Violin diagram showing the expression of Mock (blue) and zebularine treated (red) clusters. **e, f** mRNA levels of indicated genes from B16F10 (**e**) and MC38 (**f**) cells treated with DMSO or zebularine for 72 h measured by RT-qPCR. **g** mRNA levels of indicated genes from B16F10 tumor-bearing mice treated with PBS or zebularine for 12 days, measured by RT-qPCR. Data are presented as mean ± SEM of at least three independent experiments. *$p < 0.05$; **$p < 0.01$; ***$p < 0.001$ by Student's $t$-test.

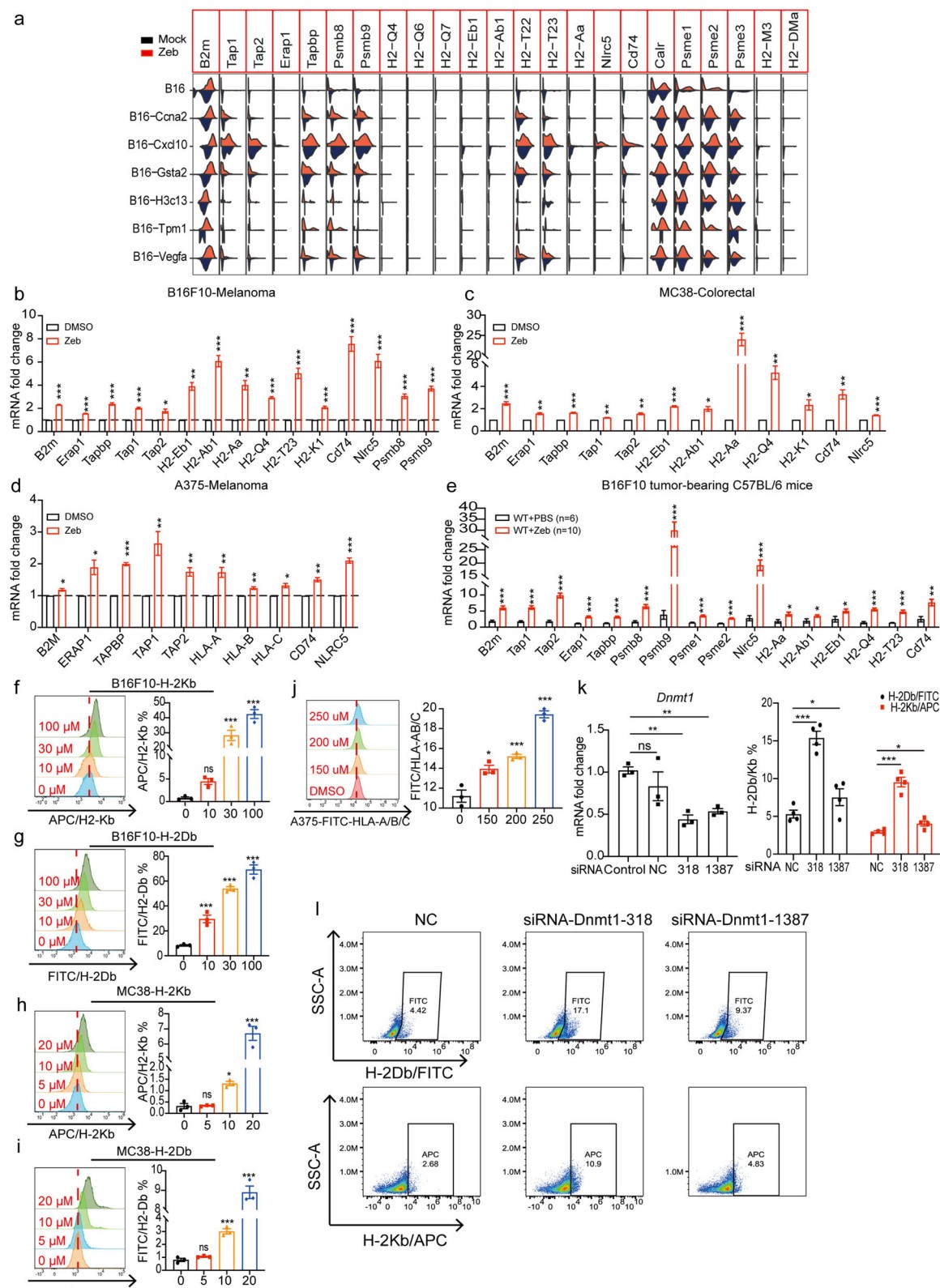

**Fig. 6 | DNMTi upregulates antigen processing and presentation. a** Violin diagram showing the expression of Mock (blue) and zebularine treated group (red) clusters. **b**, **c** mRNA levels of indicated genes from B16F10 (**b**) and MC38 (**c**) cells treated with DMSO or zebularine for 72 h measured by RT-qPCR. **d** mRNA levels of indicated genes from A375 cells treated with DMSO or zebularine for 7 days measured by RT-qPCR. **e** mRNA levels of indicated genes from B16F10 tumor-bearing mice treated with PBS or zebularine for 12 days measured by RT-qPCR. **f–i** Expression of H-2Kb/H-2Db on the surface of B16F10 (**f**, **g**) and MC38 (**h**, **i**) cells

after 72 h of zebularine treatment, detected by flow cytometry. **j** Expression of HLA-A/B/C on the surface of A375 cells after 7 days of zebularine treatment, detected by flow cytometry. **k–l** B16F10 cells were transfected with control siRNA (NC) or siRNA targeting *Dnmt1* (318 and 1387) for 3 days. RT-qPCR was used to detect the expression of *Dnmt1* (**k**) and flow cytometry was used to detect the expression of H-2Kb/H-2Db on the surface of B16F10 cells (**l**). Data are presented as mean ± SEM of at least three independent experiments. *$p < 0.05$; **$p < 0.01$; ***$p < 0.001$ by Student's $t$-test.

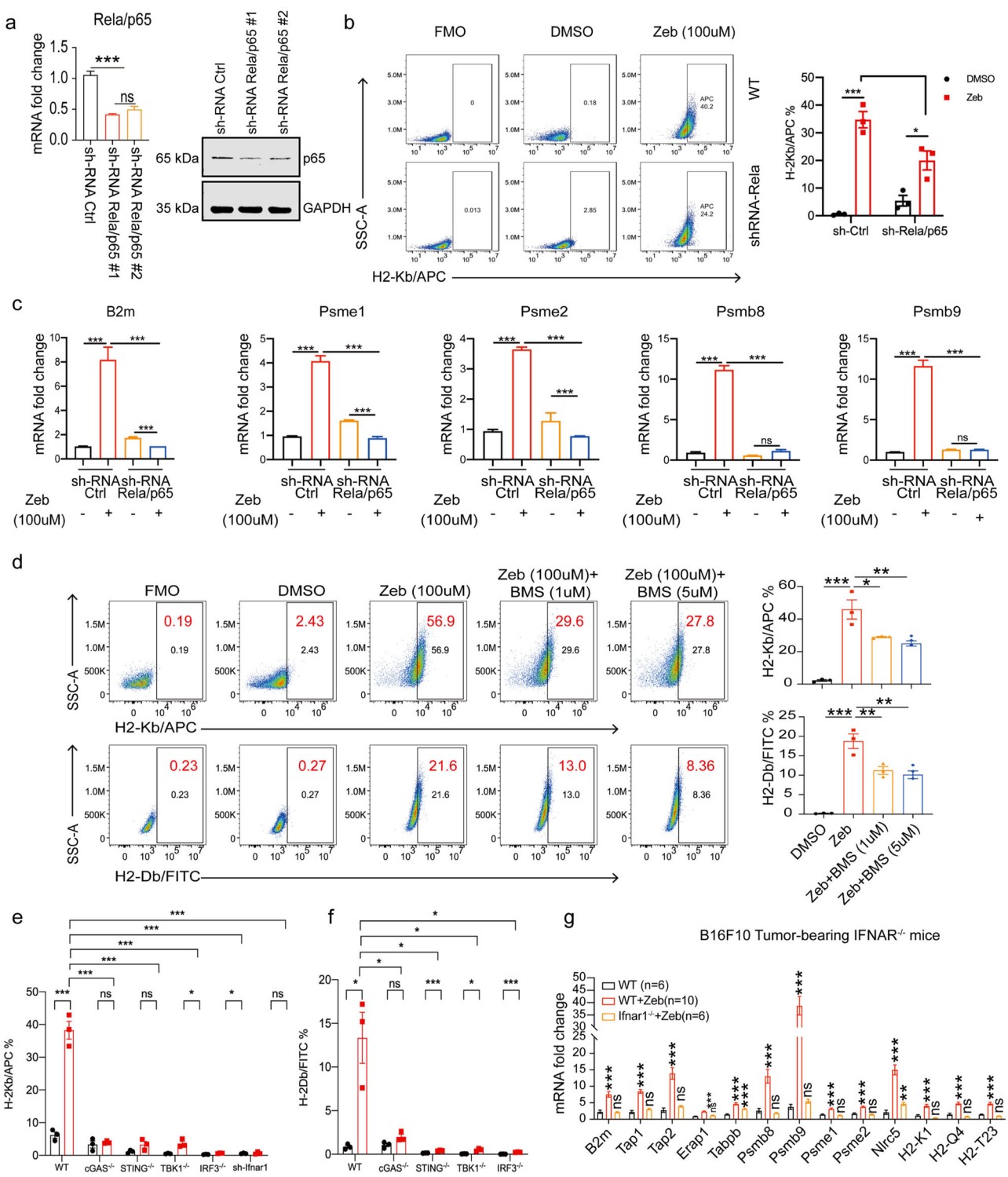

**Fig. 7 | Zebularine up-regulated antigen processing and presentation dependent on cGAS-STING-NF-κB/IFNβ signaling. a** RT-qPCR and immunoblot showing the effect of shRNA *Real/p65* knockdown in B16F10 cells. **b** Surface expression of H-2Kb in *Rela/p65* knockdown B16F10 cells 72 h after zebularine treatment, detected by flow cytometry. **c** shRNA-Ctrl or *Rela/p65*-silenced B16F10 cells were incubated with zebularine as indicated. mRNA levels of indicated gene measured by RT-qPCR. **d** Surface expression of MHC-I (H-2Kb and H-2Db) in B16F10 tumor cells treated with increasing doses of BMS-345541, a selective NF-κB inhibitor, detected by flow cytometry. **e, f** Ctrl, CRISPR-Cas9, or shRNA-silenced B16F10 cells were treated zebularine and surface MHC-I (H-2Kb and H-2Db) expression measured by flow cytometry. **g** mRNA levels of indicated genes from B16F10 tumor-bearing wild-type and *IFNAR1*[-/-] mice treated with PBS or zebularine for 12 days, measured by RT-qPCR. Data are presented as mean ± SEM of at least three independent experiments. *$p < 0.05$; **$p < 0.01$; ***$p < 0.001$ by Student's *t*-test.

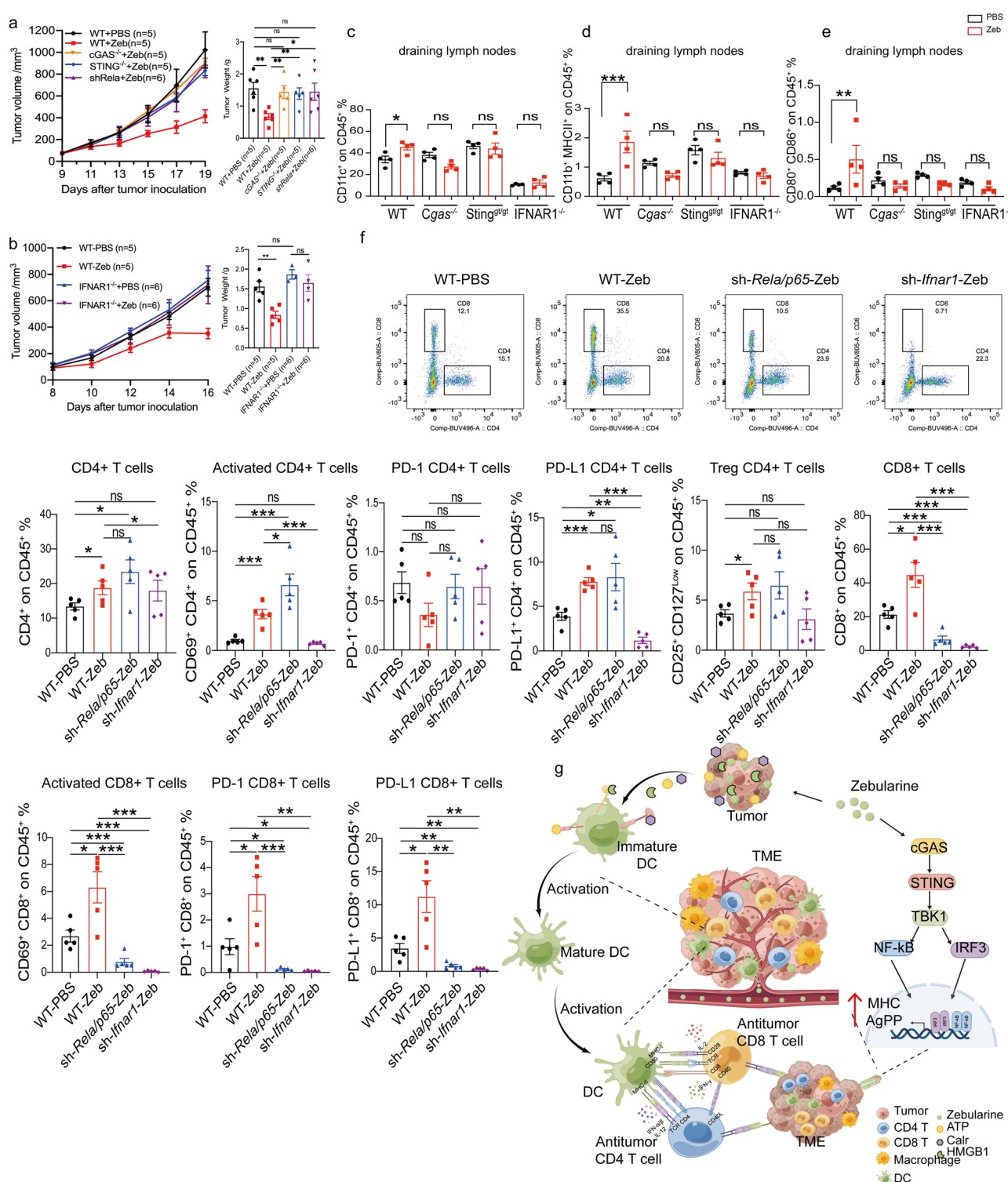

**Fig. 8 | Zebularine provokes a robust anti-tumor immune program in vivo. a** In vivo tumor growth of *Cgas⁻/⁻, Sting^gt/gt*, and sh-*Rela/p65* B16F10 tumor-bearing mice after treatment with zebularine (*n* = 5 ~ 6). **b** In vivo tumor growth of B16F10 tumor-bearing wild-type and *IFNAR1⁻/⁻* mice (*n* = 5) after treatment with zebularine. **c–e** Surface expression of CD11b⁺, CD11c⁺ MHC II⁺, and CD80⁺ CD86⁺ in the draining lymph nodes of B16F10 tumor-bearing mice detected by flow cytometry after 12 days of zebularine treatment. **f** Wild type and sh-*Rela/p65* B16F10 tumors

were implanted onto mice, and sh-*Ifnar1* B16F10 tumors were implanted onto *IFNAR1⁻/⁻* mice. Mice were subsequently mock treated or treated with zebularine for 12 days (*n* = 5). TILs were isolated for T cell analysis by flow cytometry. **g** Schematic of the signaling responses triggered by DNMTi (zebularine) chemotherapy leading to MHC-AgPPM gene induction and increased immune cell infiltration (By Figdraw, ID: OIAIR00a14). Data are presented as mean ± SEM of at least three independent experiments. *p < 0.05; **p < 0.01; ***p < 0.001 by Student's *t*-test.

is dependent on type I interferon signaling, with IFNγ further enhancing MHC-AgPPM expression.

## Zebularine provokes a robust anti-tumor immune program in vivo

To clarify the role of the cGAS-STING-NF-κB/IFNβ signaling pathway in the anti-tumor response to zebularine, we treated *Cgas*⁻/⁻, *Sting*⁻/⁻, and sh-*Rela/p65* B16F10 tumors implanted into wild-type mice, and sh-*Ifnar1* B16F10 tumors implanted into *IFNAR1*⁻/⁻ mice. Deficiency in *Cgas*, *Sting*, NF-κB/p65, or *IFNAR1* largely abolished the therapeutic effect of zebularine (Fig. 8a, b), consistent with its stimulatory effect on MHC-AgPPM gene expression. The dendritic cells and other immune cells can recognize, uptake, process, and deliver tumor antigens in the lymph node, thereby initiating an antitumor response. Accordingly, we found that zebularine promotes the maturation and activation of DCs in lymph nodes (Fig. 2g, h). As the lymph nodes play an essential role in cross-presenting antigen to T cells and initiating T cell killing of tumors, we also assessed the maturation and activation of DCs in dLNs after zebularine treatment in B16F10 tumor-bearing wild-type, *Cgas*⁻/⁻, *Sting*ᵍᵗ/ᵍᵗ, and *IFNAR1*⁻/⁻ mice. Interestingly, promotion of the maturation and activation of DCs by zebularine was impaired in dLNs of *Cgas*⁻/⁻, *Sting*ᵍᵗ/ᵍᵗ, and *IFNAR1*⁻/⁻ mice compared with wild-type mice (Fig. 8c–e and Supplementary Fig. 8a). These results further support that zebularine upregulates antigen processing and presentation, processes in which the cGAS-STING pathway and IFN play a critical role to recruit immune cells to infiltrate tumors.

As the anti-tumor effects of zebularine depend on NF-κB and type I interferon signaling pathways, we next investigated how T cells contribute to the therapeutic effects of zebularine in the TME. We harvested tumor tissue from B16F10 tumor-bearing mice for immune profiling after 12 days of zebularine treatment. FACS analyses indicated that zebularine treatment significantly increased the numbers of CD4⁺, activated CD4⁺, CD8⁺, and activated CD8⁺ T cells in tumors (Fig. 8f and Supplementary Fig. 9). Elevation of these cell subsets was abolished in *IFNAR1*⁻/⁻ mice. Moreover, we found that zebularine treatment led to a decrease in the number of CD8⁺ and activated CD8⁺ cells, but not CD4⁺ and activated CD4⁺ cells in sh-*Rela/p65*-B16F10 tumors (Fig. 8f). These results indicate that infiltration of CD8⁺ T cells into the TME in sh-*Rela/p65* tumors is impaired, which explains why zebularine had no anti-tumor effect in sh-Rela/p65 cells. In addition, we found that zebularine treatment significantly increased the expression of PD-1⁺, CD8⁺ and PD-L1⁺, and CD8⁺ cells, but these effects were abolished in sh-*Rela/p65* cells and in *IFNAR1*⁻/⁻ mice (Fig. 8f). These results indicate that zebularine induces the upregulation of PD-L1 and PD-1 expression through NF-κB and type I interferon signaling pathways; However, the specific (transcription) factors involved downstream of the NF-κB and type I interferon signaling pathways require further investigation. Together, these data suggest that the anti-tumor effect of zebularine depends at least on the co-existence of CD4⁺ and CD8⁺ T lymphocytes in the TME. Crucially, the promotion of tumor antigen processing and presentation by zebularine is dependent on NF-κB and type I interferon signaling, which ultimately affects the infiltration of T cells into the TME for killing and inhibition of tumor growth.

## Discussion

The efficacy of anti-cancer immunotherapies depends to a large extent on the TME, especially the tumor immune microenvironment. In this paper, we showed that the DNMTi zebularine induces ICD in tumor cells (Fig. 1). Zebularine treatment results in the translocation of Calr from the ER lumen to the cell membrane surface, release of ATP from tumor cells, and induction of IFN and chemokine production, which subsequently stimulates the maturation of DCs and enhances APC-driven phagocytosis and the cross-presentation of tumor antigens to T cells (Fig. 2). Thus, in tumors with insufficient DC infiltration and impaired function, addition of a DNMTi would induce the maturation of tumor-specific DCs (i.e., a DC vaccine) to achieve a stronger immune response. We also found that tumor regression mediated by zebularine was, at least in part, dependent on the co-presence of CD4⁺ and CD8⁺ T cells (Fig. 3). The clinical implication of this finding is

that the CD4:CD8 ratio should be taken into account during adoptive transfusion of CAR- or TCR-T cells as anti-cancer treatment to improve the effect of adoptive reinfusion and reduce inflammatory cytokine storms. The ability of T cells to kill tumors was enhanced in B16F10 tumor-bearing NCG mice after adoptive T cell therapy with the administration of zebularine (Fig. 3g, h). Treatment with DNA methylation inhibitors (DNMTi) results in demethylation of gene promoters and gene body regions, resulting in activation of tumor suppressors genes (p14, p15, p16)[52]. In addition, loss of tumor suppressor genes (such as Trp53, Gna13 and Cdkn1a) in the presence of an adaptive immune system relative to immunocompromised mice (SCID)[53]. Moreover, we found that zebularine treatment delayed tumor growth and reduced tumor volume only in mice with an intact immune system (Fig. 3a, b). In addition, we found that the anti-tumor effects of zebularine depend on the co-existence of CD4⁺ and CD8⁺ T lymphocytes (Fig. 3). Therefore, we speculated that the antitumor effects of DNMTi-zebularine did not have a direct effect on tumor cells, such as activation of tumor suppressor genes, but rather depends mainly on the immune system to exert antitumor effects. As a "stress response" of tumor cells, the ability of ICD to activate both innate and acquired immune responses and enhance the immunogenicity of tumor cells is of particular therapeutic interest, providing new avenues for anti-cancer immunotherapy. Thus, chemotherapeutics that induce ICD may be combined with immunotherapy to enhance the anti-tumor effects.

Clearing cancer cells involves classical adaptive immune responses mediated by APCs and CD8⁺ T cells. However, tumor cells can reduce antigen presentation and evade immune recognition by inhibiting DC function, by interfering with antigen processing and presentation to downregulate tumor cells' expression of HLA-I or MHC-I[54]. Our single-cell RNA-seq data showed that zebularine treatment of melanoma cells significantly upregulated a panel of MHC-AgPPM genes (Fig. 5a–c and Fig. 6a), innate immune-associated genes (IFN-stimulated genes, chemokines, etc.; Supplementary Fig. 10a), and genes involved in lymphocyte adhesion (such as *Vcam1*, *Icam1*, *Icam2*, and *Sele*; Supplementary Fig. 10b–d). One unexpected finding was that zebularine-induced antigen processing and presentation(MHC-AgPP) was abolished in tumor-bearing *IFNAR1*⁻/⁻ mice (Fig. 7g). As zebularine enhanced the immunogenicity of tumor cells by transcriptionally inducing MHC-AgPPM genes, which is independent of IFNγ signaling, we found that promoting MHC-AgPP is mainly dependent on type I interferon and facilitated by IFNγ. Zebularine may directly induce interferon stimulation gene (ISG) expression in tumor cells, which may enhance the capacity for antigen processing and presentation. These findings support a role for DNMTi in improving antigen presentation, promoting the absorption of tumor antigens by DCs, and cross-presenting antigens to CD8⁺ T cells, thus initiating T cell recognition and killing of tumor cells.

It was previously reported that DNMTi antitumor effects were associated with upregulation of NF-κB and IFNαβ signaling[55]. Importantly and crucially, we showed that increased expression of a set of MHC-AgPPM genes and H-2Kb/H-2Db after zebularine treatment relies on the cytoplasmic cGAS-STING DNA-sensing pathway (Fig. 7e–g and Supplementary Fig. 7a–f and Supplementary Fig. 8b, c). In contrast to previous reports, several DNMTi, including 5-azacytidine and decitabine, induce an IFN response in cancer cells by activating the dsRNA sensor[21,46]. DNMTi zebularine may trigger cytoplasmic DNA sensing, possibly by inducing ERV demethylation, expression, and reverse transcription into DNA to stimulate the cGAS-STING pathway[56]. We found that the antitumor effects of zebularine promote the processing and presentation of tumor antigen through the cytoplasmic cGAS-STING DNA-sensing pathway, thereby enhancing tumor killing by T cells. It is not clear why zebularine caused a decrease in the number of CD8⁺ and activated CD8⁺ cells, but not CD4⁺ and activated CD4⁺ cells, in sh-*Rela/p65*-B16F10 tumors. Recent evidence suggests that the major chemokines that recruit CD8⁺ T cells are those that interact with the chemokine receptor CXCR3, such as CXCL9 and CXCL10[57,58]. We found that zebularine can induce CXCL10 expression in a variety of tumor cell lines (Supplementary Fig. 1c). Thus, sh-*Rela/p65* may

impair the recruitment and infiltration of CD8[+] T cells into the TME by DNMTi treatment. Why the infiltration of CD4[+] T cells was not affected in sh-*Rela/p65* tumors requires further investigation, as does the effect of zebularine on other immune cells in the TME, such as macrophages and NK cells.

Surprisingly, we also found high expression levels of lipid metabolism genes in the B16F10 tumor cell subpopulation, including *Gsta4*, *Gsta2*, *Gstp1*, and *Enpp2* (Fig. 4b and Supplementary Fig. 4b). Lipid reprogramming plays an important role in the proliferation and migration of tumor cells, and lipid metabolites can modify the TME and affect the recruitment and function of tumor-related immune cells[59,60]. Many studies have shown that metabolic reprogramming is an important mechanism leading to tumor immune escape, and correcting the metabolic patterns of tumor cells and tumor-associated immune cells can play an important role in anti-tumor therapy[61–64]. In addition, we found that VEGF-related pathway genes showed high expression levels in melanoma cells (Fig. 4b). Therefore, an important future direction in anti-cancer therapeutics is the combination of metabolic therapy, VEGF/VEGFR inhibitor, immunotherapy and DNMTi; however, this strategy needs to consider the complexity and heterogeneity of the TME. Elucidating tumor metabolic characteristics and their interactions with the immune system is an arduous task that will need to be pursued in future studies.

In summary, our findings indicate that the DNMTi zebularine induces tumor immunogenic cell death via translocation of Calr from the ER lumen to the cell membrane surface, ATP release from tumor cells into the extracellular space, activation of APCs, enhanced phagocytosis by APCs, activation of IFN signaling, and upregulation of antigen processing and presentation genes. When activated, CD8[+] T cells specifically recognize antigenic peptides presented by the major histocompatibility complex (MHC; Vertebrate) or human leukocyte antigen (HLA; Human) Class I molecules and kill tumor cells[54]. However, tumors have developed various methods to restrict antigen HLA-I or MHC expression and evade immune recognition, such as antigen depletion, which inhibits DC function and interferes with antigen processing and presentation mechanisms. Our data suggest that DNMTi might overcome immune evasion associated with MHC downregulation and thus may be used in combination therapeutic strategies to restore the anti-tumor immune response.

Our study does show DNA methyltransferase inhibitors such as zebularine can potentiate anti-tumor immunity by inducing tumor immunogenicity and improving antigen processing through cGAS-STING pathway. Although our data show that DNMTi could promote the up-regulation of STING gene expression through demethylation function[34], whether this epigenetic regulation plays a role in altering the micro-environment by modulating specific genes is not clear and requires further investigation (such as DNA demethylation experiments). In addition, we found that the ability of T cells to kill tumors was enhanced after adoptive T cell therapy with the administration of zebularine in bearing tumor NCG mice, whether human adoptive T cell therapy pretreated with zebularine may lead to a better antitumor response in vivo also needs further investigation.

## Materials and methods
### Mouse models
Animal experiments were carried out in the C57BL/6 J and BALB/c mouse backgrounds. Female C57BL/6 and BALB/c mice at 6–8 weeks of age were purchased from Shanghai SLAC Laboratory Animal Co. (Shanghai, China). Female NOD-SCID and NCG mice, 6–8 weeks of age, were purchased from GemPharmatech Company (Jiangsu, China). B6(C)-*Cgastm1d(EUCOMM) Hmgu*/J mice (Strain: 026554), C57BL/6J-*Tmem173gt*/J mice (Strain: 017537) and B6(Cg)-*Ifnar1tm1.2Ees*/J mice (Strain: 028288) were purchased from The Jackson Laboratory (Bar Harbor, ME, USA). Mice were maintained in a Specific Pathogen-Free (SPF) animal facility at 23–25 °C and 50–60% humidity, and with 12 h light/12 h dark cycles. All experiments were conducted in accordance with the Guide for the Care and Use of Laboratory Animals approved by the Fujian Provincial Office for Managing Laboratory Animals and was guided by the Fujian Normal University Animal Care and Use Committee (Approval No. IACUC-20190004).

### Cell culture, transfection, and generation of stable cell lines
HEK293T, A375, WM-266-4, B16F10, CT26, MC38, and 4T1 cells were cultured in DMEM (Hyclone) medium supplemented with 10% FBS (Gibco), 100x Penicillin-Streptomycin (Basal Media). All cell lines were confirmed negative for mycoplasma contamination.

Cells at 60–80% confluence were transfected with the indicated constructs using Lipofectamine 2000 (Invitrogen) or Polyethylenimine (PEI, Polysciences, Sigma) in Opti-MEM medium (Gibco) according to the manufacturer's instructions. For gene knockdown or knockout, lentiviral constructs (pLKO.1-EGFP for shRNA and pLenti-V2 for sgRNA) were transfected into 293T cells together with helper plasmids (pMD2.G and psPAX2) using PEI or LipofectamineTM 2000. Media was changed 8 h post-transfection. Viral supernatants were collected at 24, 48, and 72 h post-transfection. Cells at ~50% confluence were infected with viral supernatants supplemented with 10 μg/mL polybrene (Solarbio). Following viral infection, cells were selected in the presence of puromycin or blasticidin (Gibco) for at least 3 days to generate stable cell lines and positive cells (EGFP-cells) were separated by flow cytometry. Primers are listed in Table S2.

### Compounds
Zebularine/zeb (Z4775) for use in cell experiments was purchased from Sigma-Aldrich. Gemcitabine (LY-188011), cisplatin (HY-17394), oxalipla-tin (HY-17371), and decitabine (HY-A0004) were purchased from MCE. Zebularine/zeb (350 mg/kg/d) for use in mouse experiments was provided by the Laboratory of Dr. Daliang Li and Weili Deng (Biomedical Research Center of South China, Fujian Normal University).

### Reverse transcription quantitative PCR (RT-qPCR) analysis
Total RNA was extracted from samples by RNAiso Plus (TaKaRa, Cat. No. 9108) according to the manufacturer's instructions. mRNA expression levels were quantified by quantitative real-time RT-PCR. The reverse transcription reaction was performed using 1 μg of total RNA with a HiScript II 1st Strand cDNA Synthesis Kit ( + gDNA wiper; Vazyme, Cat. No. R212). Quantitative real-time PCR was performed using ChamQ SYBR Color qPCR Master Mix (High ROX Premixed; Vazyme, Cat. No. Q441) at 95 °C for 30 s, followed by 40 cycles of 95 °C for 5 s, 55 °C for 30 s, and 72 °C for 30 s. PCR was performed on an ABI Q6 Fast Real-time PCR system (Applied Biosystems, Foster City, CA, USA), and changes in expression were calculated using the $2^{-\Delta\Delta Ct}$ method. Primers are listed in Table S2.

### Detection of the ICD biomarkers
The exposure of DAMPs (Calr, HMGB1, and ATP) of tumor cells after different treatments were detected as follows. The translocation of Calr was observed by confocal microscopy (LSM 780, Zeiss, Oberkochen, Germany) or flow cytometry (FACSymphony A5, BD Biosciences, Franklin Lakes, New Jersey, USA). Anti-Calreticulin (D3E6) rabbit mAb (Cat No. 12238) was purchased from Cell Signaling Technology and Anti-CD81 mouse mAb (Cat No. 66866-1-Ig) was purchased from Proteintech. ATP was detected using the Enhanced ATP Assay Kit (Beyotime Biotechnology, Cat No. S0027). HMGB1 was detected by confocal microscopy using anti-HMGB1 rabbit mAb (Cat No. 3935) purchased from Cell Signaling Technology, or by ELISA with a kit was purchased from Tecan, Switzerland (Cat No. ST51011), performed according to the manufacturer's instructions.

### Generation of mouse BMDCs and BMDM
BMDCs and BMDM were obtained from 6 to 8 week-old C57BL/6 female mice. Briefly, mouse femurs and tibias were flushed with cold PBS through a 70-μm cell strainer then centrifuged for 5 min at 1000 rpm. The supernatant was discarded and treated with 1 × RBC (Red Blood Cell Lysis Buffer, BD Biosciences) at room temperature for 5 min and washed twice with PBS (2% fetal bovine serum), centrifuged for 5 min at 1000 rpm, and re-suspended in RPMI 1640 medium (Hyclone). Cells (5 × 106) were seeded in 100 mm cell

culture plates in 10 mL of conditioned medium and incubated at 37 °C with 5% CO2. 75% of the medium (RMPI 1640 medium + 10% FBS) was replaced with fresh medium every 2 days, and corresponding cytokines were added as follows. BMDCs cultures: GM-CSF (20 ng/mL) and IL-4 (10 ng/mL); BMDM cultures: M-CSF (20 ng/mL).

### Tumor cell phagocytosis assay
Carboxyfluorescein succinimidyl ester (CFSE; BD Pharmingen) labeled tumor cells were plated at $1 \times 10^6$ cells/well and allowed to adhere for 3 h in treated 6-well culture plates. Cells were treated with zebularine for 72 h. $1 \times 106$ eFluor®670 (eBioscience) phagocytes (BMDCs or BMDM) were co-cultured with the treated tumor cells for 4 h at 37 °C. Cells were collected on ice, washed 2× with cold PBS (2% fetal bovine serum), and suspended in PBS containing 2% FBS. Cells were analyzed and data were acquired on BD FACSymphony A5, and FACS-Diva 7 software following the standard gating strategy for flow cytometry analysis. Phagocytosis was analyzed using FlowJo V10.6 software and calculated as the percentage of CFSE$^+$ cells within the total eFluor®670+ phagocyte population[65].

### Tumor growth and treatments
We used the gold standard assessment of immunogenic cell death in oncological mouse models, as described previously[36]. For the immunization study, CT26 tumor cells were pretreated with zebularine (150 μM), deci-tabine (100 μM), oxaliplatin (100 μM), platinum (30 μM), or gemcitabine (10 μM) for 3 days followed by subcutaneous inoculation into BALB/c mice as a vaccine, respectively. After 8 days, mice were re-challenged with live CT26 cells. Live CT26 cells were also implanted into non-immunized mice as a control. Tumor growth was monitored.

$1 \times 10^6$ B16F10, $1 \times 10^6$ MC38, or $1 \times 10^6$ CT26 cells in 100 μL PBS were subcutaneously injected into the flank of 6 to 8 week-old C57BL/6 or BALB/c, NOD-SCID female mice, respectively. After tumor cell implantation, tumor size was measured every 2–3 days by caliper until reaching 60–80 mm³, with tumor volumes calculated by the formula: length × width × height × 3.14/6. Tumor-bearing mice were pooled and randomly divided into a PBS (control) group and a zebularine treatment group. The same process was repeated for the anti-CD4 and anti-CD8 experiments, with mice injected with either $1 \times 10^6$ B16F10 or $1 \times 10^6$ MC38 cells in 100 μL PBS and tumors measured by caliper until reaching 60–80 mm³. Tumor-bearing mice were pooled and randomly divided into the following groups: (1) PBS (control); (2) PBS + Isotype; (3) zebularine treatment; (4) anti-CD4 or anti-CD8 antibody; (5) anti-CD4 or anti-CD8 block anti-body + zebularine treatment. Zebularine was administered starting with the first cycle of immunotherapy. Isotype controls were injected according to the same schedule. Anti-mouse CD4 and CD8 were used for depletion of T cells in immunocompetent mice. Animals were treated intraperitoneally with 200 μg of anti-CD4 or CD8 antibody per mouse and the antibody was applied every 3 days.

### Adoptive T cell transfer
Briefly, mice were euthanized and spleens were minced and passed through a 70 μm cell strainer. Erythrocytes were lysed with BD Pharm Lyse™ Lysing Buffer (Cat No. 555899) for 5 min at room temperature and washed twice with PBS (2% fetal bovine serum). Magnetic bead sorting, using a negative selection kit (Thermo Fisher Scientific), was used to sort CD3$^+$, CD4$^+$, and CD8$^+$ T cells. The purity of the enriched cells was >85%. T cells were dissolved in 200 μL of PBS and $2 \times 10^6$ cells were intravenously injected into B16F10 tumor-bearing female mice via the tail vein.

### Flow cytometry cell analysis
Mouse lymph node and tumor tissue were filtered through 70 μm strainers. Cells were incubated with Fixable Viability Stain 780 (Fvs780) for 15 min in the dark (room temperature), washed 2× with cold PBS (2% fetal bovine serum), and the Fc receptors on the cell surface were blocked with anti-CD16/32 antibody (10 μg/mL) for 15 min in the dark (4 °C). Then the cells

were stained with fluorophore-labeled antibodies for 30 min in the dark (4 °C) and then detected by flow cytometry. Data were acquired with a BD FACSymphony A5 using FACS-Diva 7 software following the standard gating strategy for flow cytometry analysis. The data was processed using FlowJo V10.6 software. The antibodies used for flow cytometry analysis are listed in Table S1.

### Single-cell RNA sequencing library generation
Droplet-based 30-end massively parallel single-cell RNA sequencing (scRNA-seq) was performed by encapsulating sorted live CD45$^-$ tumor cells into droplets. Libraries were prepared using Chromium Single Cell 30 Reagent Kits v1 according to the manufacturer's protocol (10 x Genomics). The generated scRNA-seq libraries were sequenced using an Illumina NovaSeq.

### Single-cell RNA-seq processing
In this study, single-cell RNA-seq data was processed using Cell Ranger (v4.0.0) from the 10 x Genomics platform, generating downstream data for subsequent analyses. All samples were imported into Seurat (R package v4.2.0) and merged, with samples exhibiting mitochondrial gene expression >10% and gene counts of >6000 or <500 being discarded. The resulting count matrix data was normalized, scaled, and clustered based on the top principal components of genes with high variability. Seurat's t-SNE algorithm generated t-SNE plots and visualized gene features in resulting cell clusters. To identify cell-cluster markers, Seurat's FindMarker algorithm was utilized, which subsequently enabled the generation of volcano plots for the respective groups.

### Single-cell RNA-seq result analysis
In our experiment, we applied unsupervised clustering analysis to categorize cell types and subpopulations in scRNA-seq data. Initially, we identified four subpopulations based on marker genes reported in the article, including B16 cells (Dct, Pmel, Tyrp1, Mlana), CAF cells (Col1a1, Col3a1, Col5a1, Fstl1, Bgn), macrophage cells (Cd68, Adgre1, Itgam, Csf1r), and T cells (Cd3e, Cd2, Cd3d, Cd3g)[44]. The remaining B16 subgroups were divided based on the functional enrichment of top genes. These subgroups include: B16-Vegfa (*Bnip3*, *Car9*, *Vegfa* and *Slc2a1*), B16-Cxcl10 (*Ifit3b*, *Tgtp1*, *Ifit1* and *Isg15*), B16-Ccna2 (*Cenpf*, *Ccna2*, *Cdca8* and *Cdk1*), B16-Gsta2 (*Gsta4*, *Gstp1*, *Gsta2* and *Enpp2*), B16-Tpm1 (*Tpm1*, *Map1b*, *Sema6b*, *Fbln1* and *Sema6d.1*), B16-H3c13 (*Hist1h2af*, *Hist1h1d*, *Hist1h2ac*, *Hist2h3b* and *Chd2*). These subgroups were named according to the most representative gene in each group.

### Pathway analysis and functional annotation
We used Gene Ontology enrichment analysis and Single-sample Gene Set Enrichment Analysis (ssGSEA) for functional analysis. Gene signatures scores of samples were evaluated using R package GSVA. GO and KEGG analyses were performed by applying the "clusterProfiler" package.

### Immunoblotting
The western blot bands exposed above were all derived from the same PVDF membrane, which we trimmed according to the protein Maker as well as the molecular weight of the target protein at the end of transmembrane and blocking. Then, they were detected using the corresponding antibodies respectively.

### Statistical analysis
Statistical analyses were performed using GraphPad Prism 8 software. The quantitative data are presented as mean ± SEM from at least three independent experiments. *$p < 0.05$; **$p < 0.01$; ***$p < 0.001$ by Student's *t*-test.

### Reporting summary
Further information on research design is available in the Nature Portfolio Reporting Summary linked to this article.

## Data availability

The raw sequence data (Single cell RNA-Seq) reported in this paper have been deposited in the China National Center for Bioinformation, Chinese Academy of Sciences (CRA010462) that are publicly accessible at https://ngdc.cncb.ac.cn/gsa. Uncropped and unedited Western blots are provided in Supplementary Fig. S11. The source data behind the graphs in the paper can be found in Supplementary Data file. Schematic drawn by Figdraw (account number: 514147045184253952) or PowerPoint. All other data are available upon reasonable request for Lead Contact/corresponding author.

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

## Acknowledgements

This work was supported by the National Natural Science Foundation of China (Joint Fund Project, Grant No. U22A20326), National Natural Science Foundation for Young scholars of China (Grant No. 82203439), and University-industry cooperation project of Fujian Province of China (Grant No. 2021N5003). We thank Dr. Hanze Wang of the Institute of Translational Brain Science at Fudan University for technical assistance and helpful discussions. We thank Dr Daliang Li of the Institute of Biomedical Research Center of South China, Fujian Normal University for the gift of zebularine. We thank the members of the Chen laboratories for technical assistance and helpful discussions. We also thank Dr. Stacey Tobin (The Tobin Touch Inc.) for reviewing and polishing the manuscript.

## Author contributions

Q.C. and Y.Z. designed the study. Y.Z. and K.S. performed experiments. H.Z. and J.L. analyzed the single-cell RNA-seq data. Q.C. guided and supervised the project. Y.Z. analyzed and interpreted the data. W.D. synthesized zebularine. Q.C. and Y.Z. analyzed data and wrote the manuscript.

## Competing interests

The authors declare no competing interests.
