## [Peer Review File · Communications Biology]

Reviewers' comments:

Reviewer #1 (Remarks to the Author):

This is a comprehensive study on the effects of using a DNA methyltransferase inhibitor (DNMTi) for tumor control on the immune recognition of the tumor and the anti-tumor immune response. The authors deeply investigate both intra-tumoral and TME effects of the DNMTi Zebularine. In the tumor they show it causes immunological cell death based on several markers. On the TME side, they show increased DC maturation, phagocytosis and T cell differentiation with pretreated tumor cells. They also investigated Zeb-treated mice bearing tumors, and show a T-cell-dependent anti-tumor effect of Zeb. They further investigated the mechanism of the immune boost using tumor-intrinsic single cell RNA sequencing, and identified increased antigen presentation, which they further confirmed experimentally. They then hypothesized and confirmed that increased antigen presentation and other immune effects of Zeb treatment were dependent on cGAS-STING signaling within the tumor, and consequent type-I interferon signaling events downstream of that, dependent on *Ifnar1* in the TME.

This is a comprehensive and detailed investigation that is informative and impactful for the scientific community. However, the volume of the work comes with some missing control studies (mentioned in the comments below), which makes some of the conclusions hard to establish. Besides that, I felt there was some unconvincing interpretation of the scRNA-seq data and some of the claims, which I have highlighted in the specific comments below.

Specific comments:

Lines 158-160: "We observed that upon zebularine pretreatment of tumor cells, APCs significantly promoted the differentiation of naive T cells into CD4 +or CD8 +T cells (Figures 2Q and 2R)."

- The CD8 and CD4 numbers on the FACS plot (Fig 2Q) do not seem to be in the range of the bar plots (Fig 2R). In the CT26 control, case, it looks like less than 10% of CD3+ cells are claimed to be one of CD4+ or CD8+. What are the other 90% CD3+ cells then?

192: "Distinct functional signatures of the melanoma tumor cells"

- This section mainly describes the different clusters. But it doesn't explain how these clusters differ between the control and treatment groups, which can be gleaned from the figures S4A and S4D.

Lines 215-216: "inhibited tumor cell growth, mainly by suppressing cell cycle G1 phase and expression of cell proliferation-related genes (such as *Pcna*, *Mcm6*, *Top2a* and *Mki67*) (Figure 4C, 4D and S5A)."

- Downregulation of some of these genes is not that convincing in Figure 4. Were these genes chosen based on statistical criteria or were they hand-picked for observation?

Lines 217 - 219: "...expression of *Mki67* in tumor tissues was also significantly reduced after zebularine treatments of B16F10 tumor-bearing mice compared with untreated controls, consistent with the analysis (Figure 4E), also shows that zebularine inhibits the growth of tumor..."

- This assessment seems to be based on microscopy. Controls like 'DAPI staining should have equivalent intensity' would be useful. This is likely doable with the current images.

Line 231: "Pseudo-temporal transition trajectory of melanoma tumor cells"

- The trajectory analysis does not seem to be very meaningful. Several of the clusters seem to be all over the trajectories. For example, B16-Ccna2 and -Vegfa clusters do not show that much change on Zeb treatment but the authors claim they are stalled. Furthermore, there is no mention of what's happening with the B16-Tpm1 cluster.

Lines 241-242: "We also investigated the transcriptional changes in the transition state and observed that melanoma cells could be divided into 3 subgroups (Figure S5B), the first subgroup, predominated by the B16-Ccna2 cluster, was characterized by cellular response to s..."

- This is confusing. The claim is that there are 3 subgroups of melanoma cells, but the figure shows clusters of genes and not of cells.

Lines 290-292: "We further found that small interfering RNA (siRNA) knockdown of the Dnmt1 gene encoding DNA methyltransferase promotes upregulation of the tumor cell surface antigen H-2Kb/H-2Db in B16F10 cells (Figures 6K and 6L)."

- The scRNA-seq data show upregulation of antigen presentation genes in almost all clusters. However, the FACS plots suggest a small subset of highly surface-expressing cells are increased on Zeb treatment, with no overall shift in fluorescence intensity. So I do not understand how the FACS plots are consistent with the scRNA-seq data.

Lines 300-301: "Our STRING analyses revealed interactions among MHC-AgPPM, proteasomal, and IFN-related pathways (Figure S10A)."

- Please provide appropriate references for the STRING method.

Lines 312-314: "Ablation of Rela/p65 abrogated zebularine-induced expression of surface H-2Kb/H-2Db and MHC-AgPPM genes (B2m, Psme1, Psmab8, and Psmb9; Figures 7B and 7C)."

- The gates in the FACS plot are not consistent between the WT controls and the siRNA treated samples, so the differences shown here do not seem to be correct. The result in Fig 7D seems to be appropriate though.

Lines 329-330: "In IFNAR1 ^{-/-} mice tumor models, we found that IFNAR1 depletion abrogated the anti-tumor effect of zebularine (Figure 8B)."

- A control arm of IFNAR1^{-/-} + PBS is missing. What if tumor growth is generally enhanced in IFNAR1^{-/-} mice, and Zeb treatment still reduces tumor growth to the level of the (WT + PBS) arm? A similar issue applies to Fig 8A and several other downstream analyses.

Lines 332-334: "Interestingly, we found that zebularine-induced expression of these genes was abolished in IFNAR1 ^{-/-} mice (Figure 7G)."

- It's not clear in which cells were the antigen presentation pathway genes were measured. Assuming these were measured in tumor cells, why would the depletion of IFNAR1 in the mice affect the Zeb-mediated upregulation of these genes in tumor cells? Didn't Zeb also upregulate these genes in vitro irrespective of extraneous cells carrying IFNAR1? (I see that the authors do find this as unexpected too in

the discussion, but I think it is important to clarify the presented results here in terms of what exactly we are seeing.)

Lines 353-355: "Interestingly, promotion of the maturation and activation of DCs by zebularine was impaired in dLNs of *Cgas*^{-/-}, *Sting*^{gt/gt}, and *IFNAR1*^{-/-} mice compared with wild-type mice (Figures 8C-8E and S8A)."

- There is a switch from *Cgas* or *Sting* depletion in tumor cells to their depletion in mice now. Why? The motivation and logic is not clear. Furthermore, the statistical comparisons shown in the Fig 8C-E probably need simplification to avoid confusing messages. The comparisons between Zeb-treated samples across the mouse genotypes are not appropriate, as the controls themselves seem to differ. The interpretation (that the Zeb-mediated effect is abolished in the knockouts) is valid if you just retain the within-genotype comparison to the controls.

Lines 369-373: "zebularine treatment significantly increased the expression of PD-1+, CD8 +and PD-L1+, and CD8 +cells, but these effects were abolished in *sh-Rela/p65* cells and in *IFNAR1*^{-/-} mice (Figure 8F). These results indicate that zebularine induces the upregulation of PD-L1 and PD-1 expression through NF-κB and type I interferon signaling pathways; however, the specific transcription factors involved need to be further studied."

- This is a confusing claim. First of all, I think *IFNAR1*^{-/-} mice were not used to investigate these cell types, rather *sh-Ifnar1* tumor cells were used. If so, this claim doesn't make sense. Secondly, PD-L1 induction is supposedly in the T cells while the NF-κB pathway is implied to be relevant in the tumor cells, so why are the authors talking about downstream transcription factors for PD-L1 induction?

(Differential expression analysis)

- I appreciate the efforts to comprehensively show the status of the called genes in the different clusters. But the volcano plot seems strange. Are the q-values really that good (reaching 10⁻³⁰⁰ for the top hits)? Was this a pseudobulk analysis? More details would be appreciated.

Fig S7F: It states "B16F10 tumor-bearing wild type /*Cgas*^{-/-} /*Sting* ^{gt/gt} mice", but are the mice of those genotypes or the tumor cells?

Fig S8C: The gates are not aligned between the vertical panels. Especially, *sh-Ifnar1* is quite misaligned, and even in the controls there seems to be a right-shift. There is no discussion on the observation.

Reviewer #2 (Remarks to the Author):

This study is a very complex and detailed analysis describing a number of mechanisms by which DNMTi inhibitor zebularine can activate antitumor immune responses. It represents a continuation of

the recent article published by the same group (Lai J et al., 2021). Numerous in vitro, as well as in vivo experiments have been performed and the data are well documented. The findings document immunogenic impacts on tumor cells, increased antigen presentation and T cell activation with a special attention paid on stimulation of cGAS-STING-NF-kappaB/IFN β signaling to enhance tumor cells immunogenicity. The results are rather confirmatory and support previous studies. Indeed, immunological impacts of DNMTis, such as induction of antigen presentation, induction of MHC and costimulatory molecules, dendritic cells activation etc., has already been intensively studied previously and also activation of the cGAS/STING pathway probably due to the ERVs demethylation has been documented (Chiappinelli KB et al., 2015, cited in this manuscript). However, this study due to its complexity brings new data and is of interest for experts in the field. Especially in vivo studies documenting the importance of the immune system for the zebularine antitumor effects are very important. The limitation of the study is that no DNA methylation studies have been performed, so it is not possible to link observed effects of zebularine to demethylation of particular genes and other mechanisms, such as the genotoxic effects of the compound can be responsible. There is only indirect evidence from experiments using downregulation of DNMT-1 with siRNA that the observed effects are really epigenetic based on DNA demethylation.

Specific comments:

- The title of the manuscript seems to be confusing since no direct evidence that the effects observed were really related to demethylation of particular genes was brought.
- DNA demethylation experiments should be completed or the manuscript Discussion should be modified, limitations of the study mentioned, and more citations included.
- Do you have any data documenting the impacts of zebularine on immunosuppressive cells populations, such as MDSCs or Tregs?
- The fact that zebularine had no antitumor effects in immunocompromised mice is quite surprising because it suggests that direct effects on tumor cells, such as activation of tumor suppressor genes did not take place. Can you discuss this point more in detail?

Dear Editors and Reviewers:

Thank you for your constructive comments on my manuscript. Please find enclosed our revised manuscript “Zebularine potentiates anti-tumor immunity by inducing tumor immunogenicity and improving antigen processing through cGAS-STING pathway” (Original title: “Inhibition of DNA methylation potentiates anti-tumor immunity by inducing tumor immunogenicity and improving antigen processing”. ID: COMMSBIO-23-2801-T) for the Communications Biology. We have addressed all concerns raised by the reviewers and have extensively modified the manuscript. All major changes are highlighted in red in the revised manuscript and our specific responses are:

Reviewer #1:

1. Lines 158-160: "We observed that upon zebularine pretreatment of tumor cells, APCs significantly promoted the differentiation of naive T cells into CD4 +or CD8 +T cells (Figures 2Q and 2R)."

- The CD8 and CD4 numbers on the FACS plot (Fig 2Q) do not seem to be in the range of the bar plots (Fig 2R). In the CT26 control, case, it looks like less than 10% of CD3+ cells are claimed to be one of CD4+ or CD8+. What are the other 90% CD3+ cells then?

Response:

We were really sorry for our careless mistakes. We have corrected the bar plots for Fig 2R in the revised manuscript (Fig 2R); We use the Down Sample plugin (Flow Jo) to set the number of cells in each group to the same level, and then perform gating CD4+ and CD8+. We have put the FACS gating strategy in the supplementary diagram (Figure S2D).

In the CT26 control, we repeated the experiment three times, and the result showed that the proportion of CD4 and CD8 cells was less than 10%. We can observe from Fig 2L that when B16F10 or CT26 is co-cultured with BMDCs, the phagocytic ability of BMDCs towards B16F10 is greater than that of CT26. Thus, we consider the CD4⁺CD8⁻ cell subsets mainly include tumor cells, BMDCs and un-differentiated naive T cells. We also found the different ability of B16F10 or CT26 to activate BMDCs and stimulate T cell proliferation and differentiation.

2. Lines 192: "Distinct functional signatures of the melanoma tumor cells".

- This section mainly describes the different clusters. But it doesn't explain how these clusters differ between the control and treatment groups, which can be gleaned from the Figures S4A and S4D.

Response:

We very much appreciate the reviewer for the important comments. As suggested by the reviewer, we make changes in the revised manuscript which are also shown in the following **(Lines 222-227)**:

From these clusters, we found that the number of B16-Ccna2, B16-Cxcl10 and B16-Tpm1 clusters was increased, and the number of B16-Vegfa, B16-Gsta2 clusters was decreased in the zebularine group compared with the mock group. These results indicated that zebularine treated tumor cells may promote inflammatory response, inhibit cell proliferation and reduce cell energy metabolism (S-Figure 4A and 4D).

3. Lines 215-216: "inhibited tumor cell growth, mainly by suppressing cell cycle G1 phase and expression of cell proliferation-related genes (such as Pcna, Mcm6, Top2a and Mki67) (Figure 4C, 4D and S5A)."

- Downregulation of some of these genes is not that convincing in Figure 4. Were these genes chosen based on statistical criteria or were they hand-picked for observation?

Response:

Thank you for this valuable feedback. Selection of these genes (MCMP, Ki-67 and PCNA), is based on common markers used in the literature to observe cell proliferation which are widely used in clinical applications and research (such as, PMID: 27246286, PMID: 27246286). In single-cell analysis we observed down-regulation of the expression of these genes in the zebularine group relative to the mock group. At the same time, the expression of Mki67 in tumor tissues was also significantly reduced after zebularine treatments of B16F10 tumor-bearing mice compared with untreated controls, consistent with the single-cell analysis (Fig 4E).

4. Lines 217 - 219: "...expression of Mki67 in tumor tissues was also significantly reduced after zebularine treatments of B16F10 tumor-bearing mice compared with untreated controls, consistent with the analysis (Figure 4E), also shows that zebularine inhibits the growth of tumor..."

- This assessment seems to be based on microscopy. Controls like 'DAPI staining should have equivalent intensity' would be useful. This is likely doable with the current images.

Response:

We sincerely appreciate the valuable comments. We re-adjusted the DAPI staining so that the DAPI staining had equal intensity. These changes will not influence results of the paper. The changes were made in **Figure 4E** corrections in the revised manuscripts.

5. Line 231: "Pseudo-temporal transition trajectory of melanoma tumor cells"

- The trajectory analysis does not seem to be very meaningful. Several of the clusters seem to be all over the trajectories. For example, B16-Ccna2 and -Vegfa clusters do not show that much change on zebularine treatment but the authors claim they are stalled. Furthermore, there is no mention of what's happening with the B16-Tpm1 cluster.

Response:

We very much appreciate the reviewer for the important comments. Since the pseudo-temporal transition trajectory analysis has no impact on the conclusions of the full text, accordingly, we have deleted this section in the revised manuscript --"Pseudo-temporal transition trajectory of melanoma tumor cells". These changes will not influence the content and framework of the paper.

6. Lines 241-242: "We also investigated the transcriptional changes in the transition state and observed that melanoma cells could be divided into 3 subgroups (Figure S5B), the first subgroup, predominated by the B16-Ccna2 cluster, was characterized by cellular The author' s answer to s..."

- This is confusing. The claim is that there are 3 subgroups of melanoma cells, but the figure shows clusters of genes and not of cells.

Response:

We very much appreciate the reviewer for the important comments. The above analytical results are further obtained by the proposed time series analysis. Because of this, the reviewer's suggestion to delete this part of the proposed timing analysis was agreed to in question-5. To avoid further misunderstanding to the readers, we have deleted the content of the proposed time series analysis. These changes will not influence the content and framework of the paper.

7. Lines 290-292: "We further found that small interfering RNA (siRNA) knockdown of the Dnmt1 gene encoding DNA methyltransferase promotes upregulation of the tumor cell surface antigen H-2Kb/H-2Db in B16F10 cells (Figures 6K and 6L)."

- The scRNA-seq data show upregulation of antigen presentation genes in almost all clusters. However, the FACS plots suggest a small subset of highly surface-expressing cells are increased on Zebularine treatment, with no overall shift in fluorescence intensity. So I do not understand how the FACS plots are consistent with the scRNA-seq data.

Response:

Thank you for your question. FACS plots showed that a small fraction of highly surface-expressing cells increased after siRNA DNMT1 treatment. We think that this may be related to both the amount and efficiency of transfected siRNA. Meanwhile, in vitro (Figure 6B-6C and Figures 6F-6I) and in vivo (Figure 6E) experiments, we obtained the same results by DNA methylation inhibitor-zebularine. These results indicate that inhibition of DNA methylation, either by knocking down DNA methylation transferase expression or

by treatment with zebularine, promotes the processing and presentation of tumor cell antigens consistent with the scRNA-seq data.

8. Lines 300-301: "Our STRING analyses revealed interactions among MHC-AgPPM, proteasomal, and IFN-related pathways (Figure S10A)."

- Please provide appropriate references for the STRING method.

Response:

As suggested by the reviewer, we have added references in the revised manuscript (Line 278). -- [47] "Szklarczyk D, et al. The STRING database in 2023: protein-protein association networks and functional enrichment analyses for any sequenced genome of interest. *Nucleic Acids Res.* 2023 Jan 6;51(D1):D638-D646. doi: 10.1093/nar/gkac1000."

9. Lines 312-314: "Ablation of Rela/p65 abrogated zebularine-induced expression of surface H-2Kb/H-2Db and MHC-AgPPM genes (B2m, Psme1, Psmab8, and Psmb9; Figures 7B and 7C)."

- The gates in the FACS plot are not consistent between the WT controls and the siRNA treated samples, so the differences shown here do not seem to be correct. The result in Fig 7D seems to be appropriate though.

Response:

As suggested by the reviewer, we have re-gated FACS plot to be consistent between the WT control and the siRNA treated samples, and added the correct data in the results section (Figure 7B) in the revised manuscript. The conclusions of the reanalysis remain

unchanged.

10. Lines 329-330: "In *IFNAR1*^{-/-} mice tumor models, we found that IFNAR1 depletion abrogated the anti-tumor effect of zebularine (Figure 8B)."

- A control arm of *IFNAR1*^{-/-} + PBS is missing. What if tumor growth is generally enhanced in *IFNAR1*^{-/-} mice, and Zeb treatment still reduces tumor growth to the level of the (WT + PBS) arm? A similar issue applies to Fig 8A and several other downstream analyses.

Response:

As suggested by the reviewer, we have added relevant experimental data, including the exact location where the change can be found in the revised manuscript (Figure 8B).

This show that in the *IFNAR1*^{-/-} mouse model, tumor growth was not reduced to (WT -PBS) levels after zebularine treatment (Figure 8B). (We observed from tumor weight that the tumor weight in either *IFNAR1*^{-/-}-PBS (1.87 g) or *IFNAR1*^{-/-}-zeb (=1.65 g) were heavier than that in WT- PBS (1.56 g) and the same trend was seen in the tumor volume growth curve (Figure 8B), and also the *IFNAR1*^{-/-}-zeb group started to die while the WT- PBS group had not).

On the one hand, the large number of experimental groups (WT -PBS, WT -zeb, sh-*NF-kB*-PBS, sh-*NF-kB*-zeb, *IFNAR1*^{-/-}-PBS, *IFNAR1*^{-/-}-zeb) made it difficult to extract TILs using the Percoll method (lack of time and manpower); on the other hand, due to limited funding, there were too many experimental groups and relatively large amounts of antibodies used. Moreover, in our previous published paper (PMID: 33571681), we did not find any difference in tumor infiltration of lymphocytes (mainly CD4+ and CD8+) in the

cGAS^{-/-} and *STING*^{gt/gt} groups after treatments with PBS or zeb. Therefore, we focused on comparing the effects of zeb treatment on immune cell infiltration into the tumor tissue in WT mice, -zeb-treated mice, *sh-NF-kB*-zeb mice, and *IFNAR1*^{-/-}-zeb mice. We believe that the zeb antitumor therapeutic effect as well as the promotion of TILs into tumor tissues is dependent on the integrity of the *cGAS-STING-NF-kB/IFNAR1* pathway.

11. Lines 332-334: "Interestingly, we found that zebularine-induced expression of these genes was abolished in *IFNAR1*^{-/-} mice (Figure 7G)."

- It's not clear in which cells were the antigen presentation pathway genes were measured.

Assuming these were measured in tumor cells, why would the depletion of *IFNAR1* in the mice affect the Zeb-mediated upregulation of these genes in tumor cells? Didn't Zeb also upregulate these genes in vitro irrespective of extraneous cells carrying *IFNAR1*? (I see that the authors do find this as unexpected too in the discussion, but I think it is important to clarify the presented results here in terms of what exactly we are seeing.)

Response:

Thank you for pointing this out. We found that treatment of the cancer cell line with *IFN* β for 48 hr promoted low-level upregulation of tumor MHC-AgPPM genes (Figure S7G). In addition, in vitro (Figure 6B, 6C) and in vivo (Figure 6E) experiments, we found that zebularine-induced the expression of the tumor MHC-AgPPM genes. To exclude the possible effects of *IFN* production in vivo, we used an *IFNAR1*^{-/-} mice model to explore whether the promotional effect of zebularine on the upregulation of tumor MHC-AgPPM gene expression is dependent on *cGAS-STING-NF-kB/IFNAR1* pathway.

We very much appreciate the reviewer for the important comments. As suggested by the reviewer, we make changes in the revised manuscript which are also shown in the following (Lines 308-315):

In addition, when B16F10 cells were treated with different concentrations of IFN β alone or zebularine combined with IFN β for 2 days, RT-qPCR revealed upregulation of MHC-dependent expression of genes involved in antigen processing and presentation (Figure S7G). Moreover, we used B16F10 tumor-bearing *IFNAR1*^{-/-} mice to examine zebularine-dependent type I interferon signaling in promoting the expression of genes involved in tumor antigen processing and presentation. Interestingly, we found that zebularine-induced expression of MHC-AgPPM genes was abolished in *IFNAR1*^{-/-} mice compared to wild-type mice (Figure 7G).

12. Lines 353-355: "Interestingly, promotion of the maturation and activation of DCs by zebularine was impaired in dLNs of *Cgas*^{-/-}, *Sting*^{gt/gt}, and *IFNAR1*^{-/-} mice compared with wild-type mice (Figures 8C-8E and S8A)."

- There is a switch from *Cgas* or *Sting* depletion in tumor cells to their depletion in mice now. Why? The motivation and logic is not clear. Furthermore, the statistical comparisons shown in the Fig 8C-E probably need simplification to avoid confusing messages. The comparisons between Zeb-treated samples across the mouse genotypes are not appropriate, as the controls themselves seem to differ. The interpretation (that the Zeb-mediated effect is abolished in the knockouts) is valid if you just retain the within-genotype comparison to the controls.

Response:

As suggested by the reviewer, we make changes in the revised manuscript which are also shown in the following (Lines 332-335):

The dendritic cells and other immune cells can recognize, uptake, process, and deliver tumor antigens in the lymph node, thereby initiating an antitumor response. Accordingly, we found that zebularine promotes the maturation and activation of DCs in lymph nodes (Figure 2G-H). As the lymph nodes play an essential role in cross-presenting antigen to T cells and initiating T cell killing of tumors, we also assessed the maturation and activation of DCs in dLNs after zebularine treatment in B16F10 tumor-bearing wild-type, *Cgas*^{-/-}, *Sting*^{gt/gt}, and *IFNAR1*^{-/-} mice. Interestingly, promotion of the maturation and activation of DCs by zebularine was impaired in dLNs of *Cgas*^{-/-}, *Sting*^{gt/gt}, and *IFNAR1*^{-/-} mice compared with wild-type mice (Figures 8C-8E and S8A). These results further support that zebularine upregulates antigen processing and presentation, processes in which the cGAS-STING pathway and IFN play a critical role to recruit immune cells to infiltrate tumors.

We simplified the statistical comparisons as suggested by the reviewers (Figure 8C).

13. Lines 369-373: "zebularine treatment significantly increased the expression of PD-1+, CD8 +and PD-L1+, and CD8 +cells, but these effects were abolished in sh-Rela/p65 cells and in *IFNAR1*^{-/-} mice (Figure 8F). These results indicate that zebularine induces the upregulation of PD-L1 and PD-1 expression through NF-κB and type I interferon signaling pathways; however, the specific transcription factors involved need to be further studied."

- This is a confusing claim. First of all, I think *IFNAR1*^{-/-} mice were not used to investigate

these cell types, rather sh-Ifnar1 tumor cells were used. If so, this claim doesn't make sense. Secondly, PD-L1 induction is supposedly in the T cells while the NF-κB pathway is implied to be relevant in the tumor cells, so why are the authors talking about downstream transcription factors for PD-L1 induction?

Response:

We very much appreciate the reviewer for the important comments.

We used *sh-Ifnar1* B16F10 cells implant onto *IFNAR^{-/-}* mice (Figure legend 8F). In our revised manuscript, the content is revised (including figure legend), we have corrected the “in *IFNAR1^{-/-}* mice” into “the *sh-Ifnar1* B16F10 cells were implanted onto *IFNAR^{-/-}* mice”.

According to your suggestions, we have revised the sentence in the revised manuscript. The details are shown as follows (Lines 359-363): “These results indicate that zebularine induces the upregulation of PD-L1 and PD-1 expression through NF-κB and type I interferon signaling pathways; However, the specific (transcription) factors involved downstream of the NF-κB and type I interferon signaling pathways require further investigation”.

14. (Differential expression analysis)

- I appreciate the efforts to comprehensively show the status of the called genes in the different clusters. But the volcano plot seems strange. Are the q-values really that good (reaching 10⁻³⁰⁰ for the top hits)? Was this a pseudobulk analysis? More details would be appreciated.

Response:

We very much appreciate the reviewer for the important comments. We have checked the data again. The q-values of the single-cell differential gene analysis results were up to 10⁻³⁰⁰. We also reviewed multiple papers and found that q-values of 10⁻³⁰⁰ are common in single-cell differential gene analyses (such as, PMID: 35624211, PMID: 35413951, PMID: 28622514, and PMID: 33357445).

The analysis of single-cell differential genes was performed using the Seurat self-contained function Findmarkers. However, some people also use the pseudobulk analysis of the muscat package. According to the report, when differential gene analysis was performed with Findmarkers and pseudobulk, the up- and down-regulation of common differential genes was consistent between the two methods (PMID: 34584091).

15. Fig S7F: It states "B16F10 tumor-bearing wild type /Cgas^{-/-} /Sting^{gt/gt} mice", but are the mice of those genotypes or the tumor cells?

Response:

We sincerely thank the reviewer for careful reading. As suggested by the reviewer, we have corrected the "B16F10 tumor-bearing wild type /Cgas^{-/-} /Sting^{gt/gt} mice" into "B16F10 tumors were implanted onto wild type, Cgas^{-/-}, or Sting^{gt/gt} mice" in the revised manuscript.

16. Fig S8C: The gates are not aligned between the vertical panels. Especially, sh-Ifnar1 is quite misaligned, and even in the controls there seems to be a right-shift. There is no discussion on the observation.

Response:

We very much appreciate the reviewer for the important comments. According to your suggestions, we have carefully re-analyzed the FACS data (Fig S8C); and added the correction in the results section (Figure 7E and 7F) as well as supplemental data (Figure S8C) in the revised manuscript. The conclusions of the reanalysis remain unchanged.

Reviewer #2:

1. The title of the manuscript seems to be confusing since no direct evidence that the effects observed were really related to demethylation of particular genes was brought.

Response:

According to the reviewer's suggestion, we have changed the title to "Zebularine potentiates anti-tumor immunity by inducing tumor immunogenicity and improving antigen processing through cGAS-STING pathway" in the revised manuscript.

2. DNA demethylation experiments should be completed or the manuscript Discussion should be modified, limitations of the study mentioned, and more citations included.

Response:

We very much appreciate the reviewer for the important comments. As suggested by the reviewer, we have added limitations in the revised manuscript. The details as follows

(Lines 465-475):

Limitations of the study:

Our study does show DNA methyltransferase inhibitors such as zebularine can potentiate anti-tumor immunity by inducing tumor immunogenicity and improving antigen

processing through cGAS-STING pathway. Although our data show that DNMTi could promote the up-regulation of STING gene expression through demethylation function³⁴, whether this epigenetic regulation plays a role in altering the microenvironment by modulating specific genes is not clear and requires further investigation (such as DNA demethylation experiments). In addition, we found that the ability of T cells to kill tumors was enhanced after adoptive T cell therapy with the administration of zebularine in bearing tumor NCG mice, whether human adoptive T cell therapy pretreated with zebularine may lead to a better antitumor response in vivo also needs further investigation.

3. Do you have any data documenting the impacts of zebularine on immunosuppressive cells populations, such as MDSCs or Tregs?

Response:

Thank you for your question. The experiment result related Tregs is shown in Figure 8F. In this article, we used CD4⁺ CD25⁺ CD127^{low} to represent Tregs biomarkers (CD127 was chosen because it does not require immobilization for membrane-breaking while Foxp3 requires). We found that zebularine treatment significantly increased the expression of Tregs. Additionally, the details regarding the MDSCs cell subpopulation will be presented in another paper under preparation; we found no significant effect on infiltrating MDSCs (CD11b⁺, F4/80⁺, Ly6C⁺Ly6G⁺).

4. The fact that zebularine had no antitumor effects in immunocompromised mice is quite surprising because it suggests that direct effects on tumor cells, such as activation of tumor

suppressor genes did not take place. Can you discuss this point more in detail?

The author's answer:

As suggested by the reviewer, we have revised the description in the revised manuscript which is also shown in the following **(Lines 385-395)**:

Treatment with DNA methylation inhibitors (DNMTi) results in demethylation of gene promoters and gene body regions, resulting in activation of tumor suppressors genes (p14, p15, p16)⁵². In addition, loss of tumor suppressor genes (such as Trp53, Gna13 and Cdkn1a) in the presence of an adaptive immune system relative to immunocompromised mice (SCID)⁵³. Moreover, we found that zebularine treatment delayed tumor growth and reduced tumor volume only in mice with an intact immune system (Figure 3A and 3B). In addition, we found that the anti-tumor effects of zebularine depend on the co-existence of CD4⁺ and CD8⁺ T lymphocytes (Figure 3). Therefore, we speculated that the antitumor effects of DNMTi -zebularine did not have a direct effect on tumor cells, such as activation of tumor suppressor genes, but rather depends mainly on the immune system to exert antitumor effects.

We greatly appreciate Editor and Reviewers' hard efforts, and hope that the revision will render this manuscript acceptable for publication in the Communications Biology.

Yours Sincerely,

Qi Chen, Ph.D. Professor

Fujian Key Laboratory of Innate Immune Biology, Biomedical Research Center of
South China, College of Life Sciences, Fujian Normal University Qishan Campus 1 Keji
Road, College Town, Minhou, Fuzhou, Fujian Province 350117, China

Email: chenqi@fjnu.edu.cn

Tel: 0086-591-22868190

Reviewers' comments:

Reviewer #1 (Remarks to the Author):

I appreciate the depth of investigations the authors have performed in this study, and the corrections and additional control experiments in response to the earlier reviews. I also appreciate that some of the suggested control experiments might not be practically feasible. There are only a couple of minor clarifications I have:

Fig 2R: Thanks for correcting the barplots. The authors in their rebuttal mention that the CD8-CD4- cells may include tumor cells, BMDCs and un-differentiated naive T cells. So it looks like there is no gating on CD3, but the figure legend suggests that there is gating on CD3. Please clarify this in the legend.

Fig 4D: I am still confused about this result. The density plots do not really show a difference in expression for Mcm6 and Mki67, between the mock and Zeb groups. So why do the authors claim expression downregulation? Maybe I am missing something here?

Reviewer #2 (Remarks to the Author):

The manuscript has been thoroughly corrected according to the reviewers' comments. There are some minor points to be clarified.

- In your response to the Reviewer 1 comment 1 you state "...Thus, we consider the CD4-CD8- cell subsets mainly include tumor cells, BMDCs and un-differentiated naive T cells." So, the text "APCs significantly promoted the differentiation of naive T cells into CD4 +or CD8 +T cells" is confusing, you see the CD4 a CD8 populations enrichment and, as there was no gating for the T cells, it is not possible to make a conclusion that there was promotion of the naïve T cells differentiation.
- Fig3B please correct "immunocompomised" to "immunocompromised"
- In your response to the Reviewer 1 comment 7 you state It should be useful to present direct comparison of the effects of zebularine compared to those of siRNA on H-2Kb/H-2Db cell-surface expressions and to present the MFI data.

Dear Editors and Reviewers:

Thank you for your constructive comments and please find enclosed our revised manuscript “Zebularine potentiates anti-tumor immunity by inducing tumor immunogenicity and improving antigen processing through cGAS-STING pathway” (ID: COMMSBIO-23-2801-B). We have made all changes which are highlighted in red in the revised manuscript and our specific responses are:

Reviewer #1:

1. Fig 2R: Thanks for correcting the barplots. The authors in their rebuttal mention that the CD8-CD4⁻ cells may include tumor cells, BMDCs and undifferentiated naive T cells. So it looks like there is no gating on CD3, but the figure legend suggests that there is gating on CD3. Please clarify this in the legend.

Response:

We very much appreciate the reviewer for the comments. As suggested by the reviewer, the Figure 2(Q-R) legend has been changed from “Flow cytometry analysis of percentages of CD3⁺, CD4⁺ and CD3⁺, CD8⁺ T cells in the ternary co-incubation system” to “Flow cytometry analysis of percentages of CD4⁺ and CD8⁺ T cells in the ternary co-incubation system” (Lines 853-854).

2. Fig 4D: I am still confused about this result. The density plots do not really show a difference in expression for Mcm6 and Mki67, between the mock and

Zeb groups. So why do the authors claim expression downregulation? Maybe I am missing something here?

Response:

Sorry for the confusion, indeed the mRNA and protein levels of *Mcm6* and *Mki67* were not been consistently upregulated. To avoid this contradiction, we have made changes in the revised manuscript which are also shown in the following **(Lines 212-220)**:

Further cell cycle analysis of cancer cells showed that zebularine inhibited tumor cell growth, mainly by suppressing cell cycle G1 phase and expression of cell proliferation-related genes *Pcna*; however, we did not see the consistent change in the mRNA expression of *Mcm6* and *Mki67* by zebularine (Figure 4C, 4D and S5A). Interestingly, IHC staining showed that both *Pcna* and *Mki67* expression were significantly reduced by zebularine at the protein level compared with controls (Figure 4E and 4F). How zebularine regulates the down-regulation of *Mki67* protein expression needs to be further investigated. Thus, we suggest that zebularine inhibits the growth of tumor cells by affecting the cell cycle and cell proliferation.

Reviewer #2:

1. In your response to the Reviewer 1 comment 1 you state ...Thus, we consider the CD4-CD8- cell subsets mainly include tumor cells, BMDCs and undifferentiated naive T cells. So, the text APCs significantly promoted the

differentiation of naive T cells into CD4 +or CD8 +T cells is confusing, you see the CD4 a CD8 populations enrichment and, as there was no gating for the T cells, it is not possible to make a conclusion that there was promotion of the naive T cells differentiation.

Response:

We very much appreciate the reviewer for the comments. As suggested by the reviewer, we make changes in the revised manuscript which are also shown in the following **(Lines 156-158)**:

As shown in figure 2Q and 2R, when tumor cells were pre-treated with zebularine, both CD4+ and CD8+ T cells in the co-incubation system proliferated significantly and were more abundant than those in control groups (Figures 2Q and 2R).

2. Fig3B please correct immunocompromised to immunocompromised

Response:

Corrected (Figure 3B).

3. In your response to the Reviewer 1 comment 7 you state It should be useful to present direct comparison of the effects of zebularine compared to those of siRNA on H-2Kb/H-2Db cell-surface expressions and to present the MFI data.

Response:

Thanks for the comment.

We greatly appreciate Editor and Reviewers' hard efforts, and hope that the revision will render this manuscript acceptable for publication in the Communications Biology.

Yours Sincerely,

Qi Chen, Ph.D. Professor

Fujian Key Laboratory of Innate Immune Biology, Biomedical Research

Center of South China, College of Life Sciences, Fujian Normal University

Qishan Campus 1 Keji Road, College Town, Minhou, Fuzhou, Fujian Province

350117, China

Email: chenqi@fjnu.edu.cn

Tel: 0086-591-22868190

REVIEWERS' COMMENTS:

Reviewer #1 (Remarks to the Author):

The authors have sufficiently addressed the couple of minor comments I had earlier. The edited the text and figure legends provide appropriate clarification now.

Reviewer #2 (Remarks to the Author):

The manuscript has been corrected according to the reviewers' comments. I do not have any more comments.